# Self-Harmony: Learning to Harmonize Self-Supervision and Self-Play in Test-Time Reinforcement Learning

**Ru Wang[1], Wei Huang[2,3], Qi Cao[1], Yusuke Iwasawa[1], Yutaka Matsuo[1], Jiaxian Guo[4]**
[1] The University of Tokyo    [2] RIKEN Center for Advanced Intelligence Project
[3] The Institute of Statistical Mathematics    [4] Google Research Australia
`{ru.wang, qi.cao}@weblab.t.u-tokyo.ac.jp`
`wei.huang.vr@riken.jp   jeffguo@google.com`

## Abstract

Test-time reinforcement learning (TTRL) offers a label-free paradigm for adapting models using only synthetic signals at inference, but its success hinges on constructing reliable learning signals. Standard approaches such as majority voting often collapse to spurious yet popular answers. We introduce Self-Harmony, a framework built on a simple intuition: the correct answer should remain stable across both an original question and its paraphrase. Self-Harmony operationalizes this by employing a single model in two complementary roles: a Solver to produce answers and a Reframer to rephrase the input. Based on this, we further propose a pseudo-label method: instead of majority voting, it aggregates answer frequencies across these original and reframed views using the harmonic mean. This is a process that naturally selects for solutions stable under reframing, thereby avoiding the common trap of favoring view-dependent, spurious answers. Crucially, this requires no human supervision or auxiliary models. Across diverse reasoning benchmarks, Self-Harmony achieves state-of-the-art results at the label-free test-time setting, ranking first in 28 of 30 settings across multiple methods. Beyond accuracy, it demonstrates unprecedented robustness, with zero training failures in all experiments, underscoring its stability and reliability. Our code is publicly available at Self-Harmony.

## 1 Introduction

Scaling the reasoning capabilities of Large Language Models (LLMs) has traditionally required massive, human-curated datasets (Cobbe et al., 2021; Hendrycks et al., 2021) for supervised fine-tuning (SFT). This reliance on costly data collection has motivated a shift toward *test-time adaptation* (Sun et al., 2020; Wang et al., 2021), where models adapt "on the fly" using only unlabeled problems and available compute resources. Within this paradigm, *test-time reinforcement learning (TTRL)* has emerged as a particularly promising approach, enabling models to improve their reasoning by leveraging self-generated feedback signals without the need for external supervision (Zuo et al., 2025; Zhang et al., 2025; Prabhudesai et al., 2025; Zhao et al., 2025b).

Current TTRL approaches typically involve generating multiple solution candidates and selecting a "pseudo-label" for the model to learn from, often through self-consistency or majority voting (Wang et al., 2023; Zuo et al., 2025; Liu et al., 2025a). However, this mechanism suffers from a critical vulnerability: if the model exhibits a systematic reasoning flaw, it may produce a specific incorrect answer more frequently than the correct one. In such cases, majority voting not only fails to correct the error but can actively amplify it by selecting the flawed solution as a training target (Shi & Jin, 2025; Huang et al., 2024). This limitation highlights the need for TTRL methods that can generate robust reward signals and reliably escape the model's own "echo chamber" of errors.

To address this challenge, we introduce *Self-Harmony*, a novel TTRL framework that adapts self-play to generate reliable pseudo-labels in a fully self-contained manner. The core idea is grounded in the following intuition: **the correct answer should appear robustly across two questions that**

**are semantically equivalent but stylistically distinct, as fragile or spurious reasoning paths are often disrupted by changes in phrasing.** *Self-Harmony* operationalizes this intuition by having a single model dynamically assume two cooperative roles: a *Solver*, which generates answers to the original problem, and a *Reframer*, which rephrases the problem to provide a diverse perspective.

Building on this intuition, we observe that majority voting is not an ideal method for pseudo-label selection. Instead, we propose a pseudo-label generation strategy based on the harmonic mean of answer frequencies across both the original and reframed rollouts. **This mechanism inherently rewards answers that appear consistently in both distributions while penalizing those that arise only in one, thereby filtering out solutions driven by biased reasoning.** With the pseudo-label selected by harmonic mean, the two roles engage in cooperative self-play: the *Solver* is trained to align with the robust pseudo-label, while the *Reframer* is trained to generate informative reformulations that expose and challenge the Solver's current biases. This interaction enables the model to escape the limitations of majority voting.

We demonstrate the effectiveness of *Self-Harmony* through extensive experiments on multiple reasoning benchmarks using open-source models ranging from 1.7B to 8B parameters. Our contributions are threefold: (1) a novel TTRL framework, *Self-Harmony*, that adapts self-play to improve reasoning without labels or external models; (2) a robust pseudo-label selection mechanism based on the harmonic mean, which mitigates common failure modes of majority voting; and (3) state-of-the-art results across five open-source models and six challenging reasoning datasets. Notably, with only 16 original and 16 reframed rollouts, *Self-Harmony* boosts the performance of Llama-3.1-8B (Grattafiori et al., 2024) on GSM8K (Cobbe et al., 2021) from 60.5% to 91.6%, and improves Qwen3-4B (Yang et al., 2025) on MATH500 (Lightman et al., 2024) from 60.2% to 78.5%.

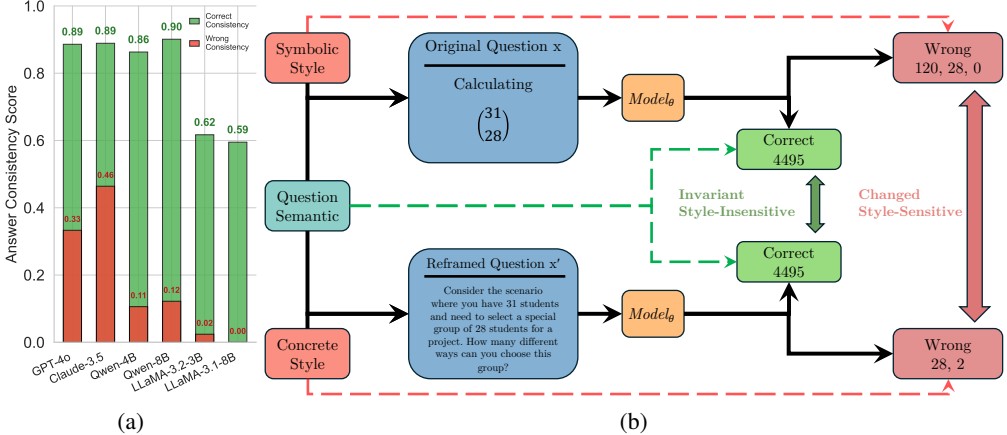

(a)    (b)

Figure 1: (a) The illustration of correct answer consistency and wrong answer consistency of some Large Language models in the original question and semantic reframed question pair. We could see the correct answer consistency is significantly higher than the wrong answer consistency, which means if models answer the original question correctly, then it will be more likely to keep the same answer asked by a reframed question. (b) Illustration showing that the hypothesis that how question description style can influence model outputs. The original question $x$ and its faithfully paraphrased counterpart $x'$ share the same underlying semantic meaning but differ in surface style. A single model $Model_\theta$ answers both questions, producing either correct or incorrect responses. **This highlights the hypothesis that correct answers should remain stable across paraphrased views, while spurious errors are more dependent on stylistic variations.**

## 2 RELATED WORK

### 2.1 SELF-IMPROVEMENT FOR LLM REASONING AT TEST TIME

A central challenge in advancing LLM reasoning is reducing reliance on large-scale supervised datasets. Test-Time Reinforcement Learning (TTRL) has emerged as a promising paradigm, enabling models to adapt during inference using only unlabeled problems (Zuo et al., 2025; Zhang et al., 2025; Prabhudesai et al., 2025; Zhao et al., 2025b). Many TTRL methods employ a rule-based verifier to provide learning signals. The most common intrinsic verifier is majority voting

(or self-consistency) (Zuo et al., 2025; Liu et al., 2025a), where the most frequent answer across multiple rollouts is selected as a pseudo-label for fine-tuning. While effective in some cases, this approach can amplify systematic biases: if a flawed reasoning process consistently produces the same incorrect answer, majority voting will reinforce that error with high confidence. Other methods mitigate this issue by leveraging powerful external models for verification (Lightman et al., 2024; Khalifa et al., 2025; Zhao et al., 2025a), or employing external reward models to guide reinforcement learning (Welleck et al., 2024; Lambert et al., 2024). However, such approaches violate the principle of a fully self-contained, test-time setting. In contrast, *Self-Harmony* operates as a pure TTRL framework, directly addressing the pseudo-label selection problem without relying on external supervision.

## 2.2 Self-Play and Data Augmentation

Self-play has been successfully adapted to LLMs, primarily in adversarial contexts where agents compete to identify and exploit vulnerabilities (e.g., "red-teaming") (Perez et al., 2022; Ge et al., 2024; Kuba et al., 2025). In contrast, our work introduces a novel cooperative self-play framework. Rather than competing, a single model assumes two collaborative roles—a *Solver* and a *Reframer*—that work together to produce solutions robust to changes in problem formulation (Xi et al., 2023; Zhou et al., 2024; Chen et al., 2024; Liu et al., 2024; Liang et al., 2025; Zhang et al., 2025). This reframing mechanism can be viewed as a form of online data augmentation, where the objective is to generate diverse perspectives of a problem. The key intuition is that correct answers should remain invariant under such reformulations, whereas incorrect answers arising from fragile reasoning patterns will not be consistently reproduced. This cooperative dynamic—aimed at distilling stable truths rather than exploiting weaknesses—distinguishes our approach from prior adversarial self-play methods (Perez et al., 2022; Ge et al., 2024).

## 2.3 Unsupervised Signal Selection by Enforcing Consistency Across Views

At its core, *Self-Harmony* is an unsupervised method that constructs its own supervisory signal without relying on ground-truth labels. Recent work uses self-consistency as an unsupervised signal in reasoning and preference learning (Prasad et al., 2024; Huang et al., 2022), but these methods enforce consistency within a single view—typically via majority voting—whereas we seek a stronger criterion that goes beyond simple vote accumulation. Our pseudo-label selection mechanism is inspired by the information maximization (Infomax) principle, which encourages representations that remain invariant across different "views" of the data (Linsker, 1988; Bell & Sejnowski, 1995; Hjelm et al., 2019; Tian et al., 2020). This approach mirrors the principles of Invariant Risk Minimization (Arjovsky et al., 2019) and Group-Invariant Learning (Chen et al., 2020) by treating paraphrases as distinct environments, ensuring that selected answers are robust to surface-level transformations rather than relying on view-dependent spurious correlations. In our setting, the *Solver* and *Reframer* produce solutions to the original and reformulated problems, thereby creating two distinct distributional views. To enforce agreement across these views, we employ the harmonic mean of answer frequencies, which provides a stringent yet practical criterion (Xie et al., 2020; Sohn et al., 2020). Because the harmonic mean heavily penalizes low values, an answer must appear frequently in both distributions to be selected. This effectively filters out spurious solutions that arise from a single fragile reasoning path but fail to generalize under reformulation, enabling the model to construct a reliable, unsupervised learning signal.

## 3 Method

Adapting a pre-trained LLM to a targeted test set without ground-truth labels at test time requires the model to generate and refine its own training signal from uncertain rollouts. In this setting, pseudo-label selection becomes the bottleneck. To motivate our approach, we first formalize the problem and illustrate the limitations of prevailing pseudo-labeling strategies, and then introduce our core mechanism for generating dual-view consistency: using the model itself to paraphrase the original question and create a new, diverse problem perspective. Building on this, we provide a theoretical analysis showing that harmonic mean consensus is a better pseudo-labeling under multi-view consistency. Finally, we introduce *Self-Harmony*, a practical algorithm that operationalizes this principle through structured self-play and robust policy optimization.

### 3.1 PROBLEM FORMULATION AND MOTIVATION

Given an unlabeled test dataset, for each question $x$ in the set, we aim to adapt our pre-trained LLM $\pi_\theta$ to get a better performance on that test dataset. To achieve this, the model generates $n$ rollouts for the input question $x$. Since ground-truth labels are unavailable during test-time adaptation, a pseudo-label $y^\star$ must be inferred from the rollouts and used as feedback for policy optimization. The quality of this pseudo-label directly dictates the stability of the adaptation process.

A dominant approach is *majority voting*: the answer most frequently generated by the model is selected. However, this simplicity comes at a cost. When $p(\text{Correct} \mid x) < p(\text{Wrong} \mid x)$, Liu et al. (2025b) formally shows that as we draw more samples, the chance that majority voting recovers the correct answer actually converges to zero — meaning that majority voting amplifies the wrong solution instead of correcting it. To resolve the "majority-vote trap" in LLM test time adaptation, we draw inspiration from a common human robustness heuristic: when confronted with uncertainty, people often check solutions across multiple perspectives or reformulate the problem to verify consistency (Pólya, 1945; Spiro et al., 1988). We hypothesize that **the correct answer, even if not the most popular, should appear robustly across different, faithfully paraphrased versions of the question.** Inspired by this, we argue that pseudo-labels should not be judged solely by popularity within a single question description, but rather by *invariance across multiple views*. This motivates an alternative criterion based on information-theoretic principles.

### 3.2 THEORETICAL ANALYSIS: FROM MAJORITY VOTING TO HARMONIC MEAN

Our analysis begins by showing how a simple extension of majority voting to multiple views can still fail. We then introduce an information-theoretic objective grounded in a view-invariance assumption. We demonstrate that the *harmonic mean* emerges as a principled second-order approximation of this objective, providing a robust mechanism for selecting pseudo-labels.

Building on prior research in dual-view approaches (Zhang et al., 2025; Federici et al., 2020), we extend the idea of majority voting to a multi-view setting. Let $p_0(a)$ be the probability that the policy $\pi_\theta$ generates answer $a$ for the original query $x$, and $p_1(a)$ be the probability that the policy $\pi_\theta$ generates answer $a$ for a reformulated query $x'$. A straightforward extension of the single-view majority voting principle is to aggregate the evidence from both views. This approach, which we term *dual-view majority voting*, selects the answer with the highest combined probability: $y^\star = \arg\max_a \big( p_0(a) + p_1(a) \big)$, This method can resolve some single-view failures. For instance, if the correct answer $C$ is less probable than a wrong answer $W$ in the first view ($p_0(C) < p_0(W)$) but more probable in the second ($p_1(C) > p_1(W)$), their sum may correctly favor $C$. However, dual-view majority voting remains vulnerable; it fails when different wrong answers dominate each view, as their summed probabilities can easily overwhelm that of the correct answer.

To overcome this limitation, we turn to an information-theoretic (Bell & Sejnowski, 1995; Tsai et al., 2020; Hjelm et al., 2019) approach guided by an assumption about the correct answers.

We view the correct answer $C$ as depending only on the underlying semantics of a query, not its superficial form. Thus, for two queries with equivalent meaning, the probability assigned to $C$ should remain unchanged. This motivates our core assumption below.

**Assumption 3.1** (View-Invariance Assumption). *For any two semantically equivalent queries $x$ and $x'$, the correct label $C$ has an approximately constant probability mass: $p(A = C \mid x) = p(A = C \mid x')$ where $A$ is the random variable of answer $a$. In contrast, the probabilities of incorrect labels $A \neq C$ vary across different views, reflecting their dependence on view-specific artifacts.*

To formalize this, we define an objective that rewards accuracy while penalizing view-dependence. Let $A$ be the random variable for the model rollout candidate answer $a$, $X \in \{x, x'\}$ the view, and $Z_a = \mathbb{I}\{A = a\}$ an indicator for whether the model's answer is $a$ or not. We adapt the standard Infomax principle, which maximizes the mutual information $I(Z_a; A)$, by introducing a penalty term $I(Z_a; X)$ that discourages dependence on the specific view. This yields the *View-Invariant Infomax* objective:

$$J_\lambda(a) = I(Z_a; A) - \lambda I(Z_a; X)$$

where the hyperparameter $\lambda$ balances the trade-off between accuracy and view-invariance.

**Theorem 3.2** (Harmonic Mean Selector from Invariant Infomax). *Assume the* view-invariance *condition (Assumption 3.1) holds. Suppose moreover that the following conditions are satisfied.*

*A1. Non-degeneracy. For every label $a$, $p_0(a) + p_1(a) < 1$. (This simply excludes the trivial case in where the majority voting can solve.)*

*A2. Balanced–Confidence. There exists a constant $\kappa \in (0,1)$ such that for every maximiser $a^\star$ of $J_\lambda(\cdot): \left| p_0(a^\star) - p_1(a^\star) \right| \le \kappa \left[ p_0(a^\star) + p_1(a^\star) \right]$.*

*A3. Uniform View Prior. The view variable $X \in \{x, x'\}$ is assumed to be sampled uniformly, i.e., $p(X = x) = p(X = x') = \frac{1}{2}$. This assumption ensures that the penalty term $I(Z_a; X)$ reflects dependence on the view itself rather than bias in the sampling distribution.*

*Then, for the penalty weight $\lambda = 2$, the pseudo label that maximises the second-order approximation of the* view-invariant Infomax *objective $J_2(a) = I(Z_a; A) - 2\,I(Z_a; X)$ is obtained by the harmonic mean of the two view-probabilities:*

$$y^\star = \arg\max_a \frac{2\,p_0(a)\,p_1(a)}{p_0(a) + p_1(a)} \in \arg\max_a J_2(a)$$

The full proof can be found in Appendix G.1. Theorem 3.2 implies that the harmonic mean is not an arbitrary heuristic but a principled regularizer that enforces view-invariance. It naturally emphasizes answers that maintain consistent support across views while aggressively penalizing those that are strong in one view but weak in another. This aligns well with the intuition that a true answer should be robust to reformulation, while spurious artifacts should not.

### 3.3 *Self-Harmony* FRAMEWORK

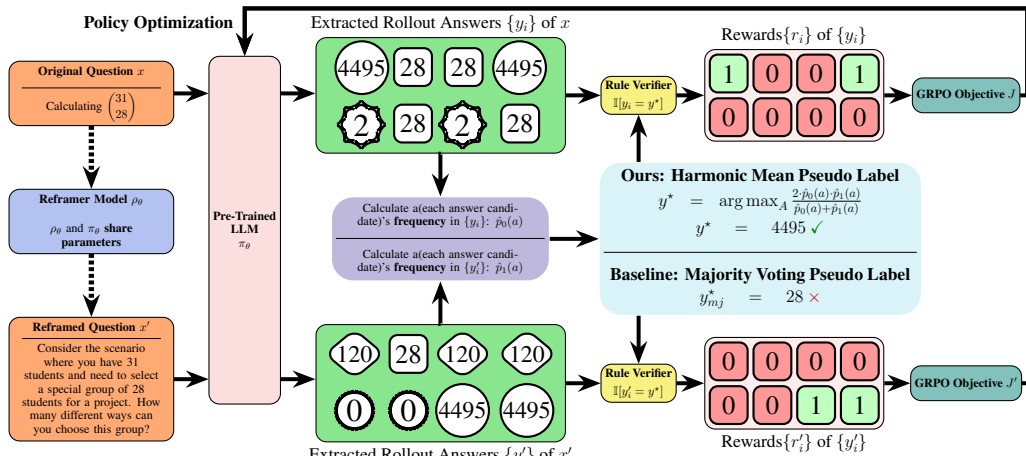

Figure 2: An overview of our framework. Given an unlabeled question $x$, the policy model $\pi_\theta$ will paraphrase it to a new question $x'$, and then generate sets of extracted rollout answers $\{y_i\}$ and $\{y_i'\}$ respectively. A rule-based verifier assigns a reward to each response based on its correctness against a target $y^\star$ got by harmonic mean instead of majority voting. These rewards are aggregated into objective functions, $J$ and $J'$, which guide the optimization of the pre-trained LLM.

Based on our theoretical analysis which concludes that View-Invariant Infomax pseudo-labeling rule for robust multi-view adaptation is to select answers based on their **harmonic mean score**, The *Self-Harmony* framework is the practical instantiation of this principle. It operationalizes this by having a single model engage in structured self-play. The model generates answers from two distinct perspectives—an original view and a self-generated, reframed view. The harmonic mean consensus rule then identifies the most consistent answer across these views to serve as the training signal, ensuring that only perspective-stable solutions are reinforced.

**Single Model, Dual Roles for View Generation.** *Self-Harmony* uses a single model ($M_\theta$) that switches between two roles via prompting. The **Solver** role $\pi_\theta$ generates answers to a query $x$, while the **Reframer** role $\rho_\theta$ creates a semantically equivalent paraphrase, $x'$. The framework follows an

intuitive "solve $\rightarrow$ reframe $\rightarrow$ solve" conceptual sequence. First, the Solver $\pi_\theta$ generates answers for $x$. Second, the Reframer $\rho_\theta$ generates $x'$. Third, the Solver $\pi_\theta$ is invoked again to generate answers for $x'$. For computational efficiency, our practical implementation optimizes this process; we detail this in the Experiments section.

**Pseudo-Label Selection via Harmonic Mean Score (HMS).** Following our theoretical analysis (Theorem 3.2), we select the pseudo-label $y^\star$ by identifying the candidate answer $a$ that maximizes the Harmonic Mean Score of its empirical support from the original $\hat{p}_0(a)$ and reframed $\hat{p}_1(a)$ views according: $y^\star = \arg\max_a \frac{2\hat{p}_0(a)\hat{p}_1(a)}{\hat{p}_0(a)+\hat{p}_1(a)}$.

## 3.4 FROM CONCEPTUAL FRAMEWORK TO PRACTICAL IMPLEMENTATION

---
**Algorithm 1** *Self-Harmony* (Conceptual Flow)

---
1: **Input:** Language Model $M_\theta$, Test Dataset $D$, Number of rollouts $N$
2: **Define:** Training Step T; Problem Solver $\pi_\theta$, Reframer $\rho_\theta$. They shared the parameters
3: **Output:** Adapted model parameters $\theta_T$
4: **for** $t = 1$ to $T$ **do**
5:     Sample minibatch $\mathcal{B} \subset D$
6:     **for** query $x \in \mathcal{B}$ **do**
7:         Generate $N$ extracted rollout answer $\{y\}$ via $y \sim \pi_\theta(\cdot \mid x)$     *// Solver on $x$*
8:         Generate $x' \sim \rho_\theta(\cdot \mid x)$     *// Reframe question from $x$ to $x'$*
9:         Generate $\{y'\}$ via $y' \sim \pi_\theta(\cdot \mid x')$     *// Solver on $x'$*
10:        Calculate A(each answer candidate)'s frequency in $\{y_i\}$: $\hat{p}_0(a)$ and in $\{y'_i\}$: $\hat{p}_1(a)$
11:        Compute the pseudo-label $y^\star$ via HMS($\hat{p}_0(a), \hat{p}_1(a)$).
12:        Compute rewards: $R_{\text{solve}}$, $R'_{\text{solve}}$, and diversity reward $R_{\text{reframe}}$
13:        Compute $\nabla J(\theta) + \nabla J'(\theta)$ via rewards
14:        Update $\theta$ by ascending $\nabla J(\theta) + \nabla J'(\theta)$
15:     **end for**
16: **end for**

---

The core principle of *Self-Harmony* is a three-step solve $\rightarrow$ reframe $\rightarrow$ solve sequence, as outlined in Algorithm 1. This conceptual flow isolates three distinct actions: an initial *solving* action on the original query, a *reframing* action, and a final *solving* action on the paraphrased query.

However, executing this sequence with three separate model calls is computationally inefficient. To create a practical and scalable algorithm, we optimize this flow by **fusing the reframing and the second solving steps into a single, structured generative action**. This is implemented in practice using a system prompt that instructs the model to first paraphrase the question and then immediately solve its own paraphrase within a single generation (see Appendix D.1 the full prompt). This optimization reduces the process to two efficient model calls: one for the initial solution and one for the joint "reframe-and-solve" trajectory.

### 3.4.1 POLICY OPTIMIZATION WITH FUSED ACTIONS

This fusion fundamentally changes the optimization problem. Instead of three actions, the policy now executes two, requiring a reward structure that reflects this new dynamic.

**Reward for the Initial Solving Action.** The first *solving* action remains unchanged. It is executed by the policy $\pi_\theta$ and rewarded for matching the pseudo-label $y^\star$, which is determined after both trajectories have been generated. The reward is simply the correctness of the generated answer $y$: $R_{\text{solve}}(y) = \mathbb{I}[y = y^\star]$.

**Reward for the Fused Reframing-and-Solving Action.** The composite action requires a reward that jointly evaluates paraphrase quality and final-answer correctness. A standard additive reward is suboptimal, as it grants partial credit for a well-formed paraphrase even when the final answer is wrong (see Appendix E.5).

We instead design a reward where answer correctness acts as a success gate. A base reward is given only when the answer is correct and is modulated by two penalty terms: a Format Penalty ($R_{\text{format}}^{\text{penalty}}$) for structural violations, and a Diversity Penalty ($R_{\text{div}}^{\text{penalty}}$), which is the Jensen-Shannon divergence between the original and reframed queries' answer. Here, each $(1 - w \cdot R^{\text{penalty}})$ term acts as a reward, so higher penalties reduce the final score. The final reward is defined as $R_{\text{fused}}(y') = (1 - w_f R_{\text{format}}^{\text{penalty}}(y'))(1 - w_d R_{\text{div}}^{\text{penalty}}(y', y))\mathbb{I}[y' = y^\star]$, where $\mathbb{I}[y' = y^\star]$ is 1 if the predicted answer

is correct and 0 otherwise. This ensures that only correct answers receive reward, while the penalties shape the signal to favor reframings that are both well-formed and meaningfully different.

Table 1: Main results on reasoning benchmarks. **Self-Harmony** consistently outperforms baseline methods across most tested models and datasets, demonstrating both superior performance and training stability. The **best** and second-best results are highlighted. Specifically, if a baseline's performance degrades significantly after its peak, we report the highest score observed across validation steps and mark it with an asterisk (*) while ours *Self-Harmony* use the score at final step.

| Methods | Mathematics | | | | Multi-Subject | Multi-Task |
|---|---|---|---|---|---|---|
| | MATH500 | GSM8K | AIME 2024 | AMC | GPQA | MMLU-Pro |
| *Qwen3-1.7B-Base* | | | | | | |
| Before RL | 42.70 | 65.58 | 3.33 | 26.50 | 20.30 | 16.61 |
| - GT-Reward | 71.80 | 85.97 | 20.83 | 53.01 | 53.80 | 85.71 |
| - Intuitor | 51.12* | 80.25* | 3.75* | 23.56* | 23.76* | 31.25* |
| - Rent | 61.08 | 78.64* | 6.45* | 32.00* | 23.47* | 18.04* |
| - Majority-Voting | 64.64 | 83.80 | 9.37 | 37.65 | 24.68 | 44.82 |
| - Co-Reward | 64.67 | 86.59 | 6.67 | 39.75 | 23.66 | 47.14 |
| - *Self-Harmony* | **69.60** | **87.47** | **10.00** | **40.51** | **27.92** | **53.66** |
| *Qwen3-4B-Base* | | | | | | |
| Before RL | 60.20 | 55.72 | 6.66 | 34.94 | 16.75 | 27.59 |
| - GT-Reward | 83.40 | 94.69 | 50.00 | 86.74 | 92.89 | 92.85 |
| - Intuitor | 72.35* | 87.53* | 10.41* | 43.59* | 32.39* | 64.55* |
| - Rent | 74.60 | 90.49* | 12.08* | 45.25* | 31.78* | 66.16* |
| - Majority-Voting | 75.75 | 93.44 | **20.00** | 49.32 | 36.51 | 52.95 |
| - Co-Reward | 76.54 | 93.47 | 12.71 | 46.98 | 24.36 | 51.79* |
| - *Self-Harmony* | **78.50** | **94.31** | **20.00** | **49.40** | **37.06** | **67.68** |
| *Qwen3-8B-Base* | | | | | | |
| Before RL | 66.80 | 84.76 | 10.00 | 45.78 | 33.44 | 50.09 |
| - GT-Reward | 84.72 | 96.51 | 56.66 | 87.95 | 93.40 | 92.85 |
| - Intuitor | 78.77* | 92.28* | 17.08* | 51.65* | 31.56* | 60.54 |
| - Rent | 77.26* | 91.20* | 18.54* | 50.82* | 37.88* | 69.91* |
| - Majority-Voting | 78.99 | 94.00 | **24.16** | 59.03 | **39.34** | 77.58 |
| - Co-Reward | 78.92 | 94.80 | 15.83 | 51.80 | 24.42 | 57.59* |
| - *Self-Harmony* | **80.00** | **95.45** | 23.33 | **59.04** | 38.07 | **77.68** |
| *Llama-3.2-3B-Instruct* | | | | | | |
| Before RL | 39.80 | 16.65 | 6.67 | 19.27 | 3.04 | 34.11 |
| - GT-Reward | 69.80 | 93.70 | 33.33 | 83.13 | 89.34 | 72.85 |
| - Intuitor | 47.78* | 16.73* | 9.16* | 23.79* | 3.23* | 34.64* |
| - Rent | 47.30* | 16.67* | 7.70* | 23.79* | 2.79* | 34.38* |
| - Majority-Voting | 46.38 | 85.98 | **13.33** | 20.48 | 22.33 | 31.43 |
| - Co-Reward | 55.22 | 89.14 | **13.33** | **25.30** | 19.79 | 34.02* |
| - *Self-Harmony* | **55.40** | **89.55** | **13.33** | **25.30** | **29.95** | **44.29** |
| *Llama-3.1-8B-Instruct* | | | | | | |
| Before RL | 41.46 | 60.48 | 3.33 | 20.48 | 14.72 | 43.75 |
| - GT-Reward | 73.60 | 95.60 | 33.33 | 80.72 | 89.84 | 91.42 |
| - Intuitor | 48.37* | 66.25* | 6.45* | 6.62* | 15.32* | 40.00* |
| - Rent | 45.98* | 69.88* | 3.95* | 21.98* | 14.56* | 40.80* |
| - Majority-Voting | 46.71* | 88.78* | 4.13 | 21.53 | 25.31 | 45.36* |
| - Co-Reward | 48.01* | 89.48 | 3.33 | 17.84 | 24.36 | 42.77* |
| - *Self-Harmony* | **50.40** | **91.59** | **10.00** | **26.51** | **28.93** | **50.00** |

## 4 EXPERIMENT

### 4.1 SETTINGS

**Benchmarks and Baselines.** We evaluate our method on a diverse set of reasoning benchmarks: MATH500 (Hendrycks et al., 2021), GSM8K (Cobbe et al., 2021), AIME 2024, AMC, GPQA-Diamond (Rein et al., 2023), and MMLU-Pro (Wang et al., 2024). Following (Zuo et al., 2025),

we report the average pass@1 over 16 rollouts. Our primary baselines are recent label-free methods: Intuitor (Zhao et al., 2025b), Rent (Prabhudesai et al., 2025), TTRL (Zuo et al., 2025), and Co-Reward (Zhang et al., 2025). For Co-Reward, we match model sizes for question rewriting rather than relying on Qwen3-32B. Specifically, Qwen3-XB-Base uses its corresponding instruct model, and Llama-Instruct uses its own. For rewriting prompts, we follow the settings described in the original papers. To ensure fairness, all methods use the same hyperparameters and official implementations.

**Models and Training Details.** To demonstrate the generality of our approach, we evaluate across a range of open-source models: Qwen3-1.7B-Base, Qwen3-4B-Base, Qwen3-8B-Base (Yang et al., 2025), Llama-3.1-8B-Instruct (Grattafiori et al., 2024), and Llama-3.2-3B-Instruct. All experiments use a learning rate of $3 \times 10^{-6}$ with a cosine schedule. For each dataset, we adopt a fully unsupervised reinforcement learning setting: a separate model is trained on the dataset without using any ground-truth labels at any stage. Although the same dataset is later used for evaluation, correctness labels are never used to generate pseudo-labels or to form any training signal, so no correctness information leaks into training. All pseudo-labels during training are produced exclusively by the model's own rollouts, and every result reported corresponds to a model independently trained on that dataset under this strictly label-free setting. A complete list of hyperparameters and experimental settings is provided in Appendix D.2.

**Fair Comparison of Computational Budget.** Multi-view methods incur additional computational cost since our framework generates answers from two views. To ensure fairness, all baselines were allocated a **comparable budget**. Specifically, if *Self-Harmony* uses $N_r$ rollouts for both the original and reframed views (totaling $2N_r$), then each baseline was also allowed $2N_r$ rollouts from its single view. This setup guarantees that performance improvements are attributable to methodological effectiveness rather than increased computation.

## 4.2 MAIN RESULTS

We evaluate *Self-Harmony* against several baseline reward modeling techniques across five reasoning benchmarks using five base models from the Qwen3 and Llama-3 series. The results are summarized in Table 1.

**Dominant Performance Across Benchmarks.** Across **30** configurations (5 models × 6 benchmarks), *Self-Harmony* ranks **first** in **28** cases and **second** in the remaining **2**, demonstrating strong generalizability across both model families (Qwen, Llama) and parameter scales (1.7B–8B). For example, on MATH500 with Qwen3-1.7B, accuracy improves from 42.70% to 69.60%, while no baseline exceeds 65%, nearly matching the oracle GT-Reward score of 71.80%.

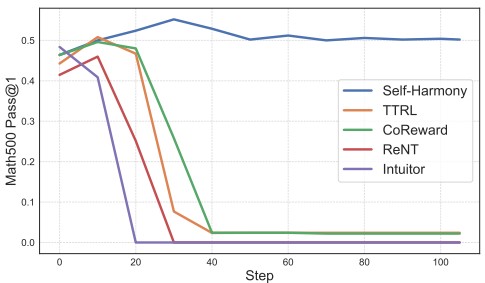
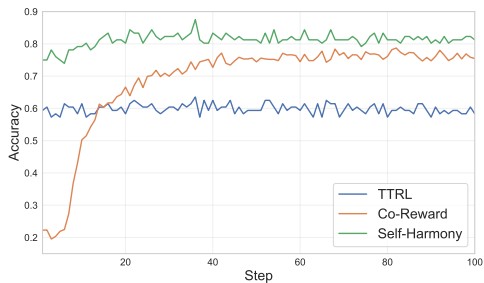

(a) Training Stability               (b) Pseudo-label accuracy on Math500-Level3

Figure 3: **(a)** *Self-Harmony* demonstrates the highest training stability with Llama-3.1-8B-Instruct on MATH500. **(b)** Comparison of pseudo-label accuracy. *Self-Harmony* consistently generates pseudo-labels with the highest accuracy throughout training, significantly outperforming Co-Reward and TTRL. For Co-Reward, we use the average accuracy for both original branch and reframed branch.

**Superior Training Stability.** As shown in Table 1, Fig. 3a, *Self-Harmony* remains stable across all experiments, highlighting the robustness of its reward modeling and training framework.

### 4.3 ANALYSIS

In this section, we conduct a detailed analysis to validate the core mechanisms of our *Self-Harmony* framework. We first analyze our Harmonic Mean Score (HMS), showing that its effectiveness in generating pseudo-labels corresponds logically with problem difficulty. We then present a comprehensive ablation study that confirms each component of our framework—including the Reframer's reward objectives and our pseudo-label selection strategy—is critical to the model's success. Collectively, these findings validate the key design principles and overall effectiveness of our approach.

#### 4.3.1 PSEUDO-LABEL QUALITY EVALUATION

The success of *Self-Harmony* is driven by the comprehensive quality of its pseudo-labels. We evaluate this quality not only through raw accuracy but also through stability and calibration metrics. As shown in Figure 3(b), our method achieves the highest accuracy on MATH500 (Level 3), stabilizing around 80–85%. To further elucidate why our approach excels, we analyze additional signal properties—including F1-score and Spearman correlation—in Appendix L. These metrics confirm that harmonic mean score provides more calibrated supervision by penalizing spurious, view-dependent answers.

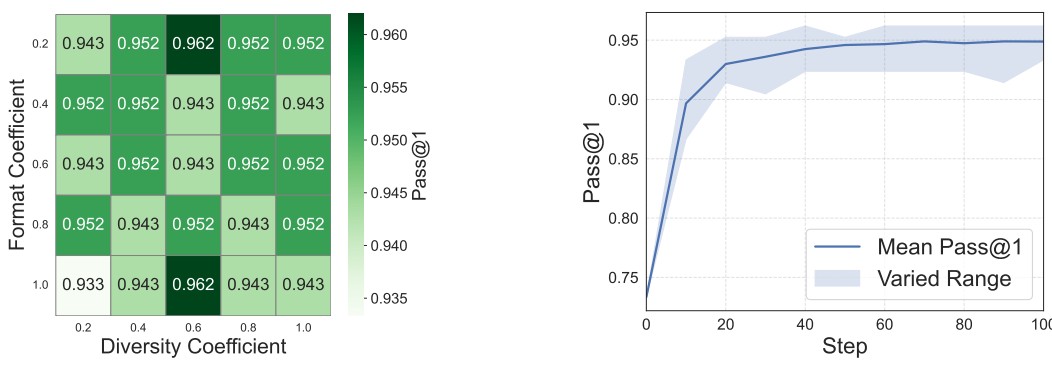

(a) Performance Heatmap         (b) Performance Range

Figure 4: Sensitivity analysis for the format ($w_f$) and diversity ($w_d$) reward coefficients. **(a)** A heatmap showing consistently high performance across a wide range of coefficient values. **(b)** A plot illustrating the uncertainty range, confirming the model's robustness to hyperparameter choices.

#### 4.3.2 MODULE ABLATION STUDY

Table 2: Ablation study on the MATH500 benchmark. We analyze the impact of removing the Format and Diversity rewards from the Reframer, and replacing our Harmonic Mean Score (HMS) based Pseudo-label Selection with simpler Majority Voting and Cross selection methods. All results are averaged over the last three evaluation checkpoints.

| Math500 | Self-Harmony (Full) | Reframer | | Pseudo-label Selection | |
|---|---|---|---|---|---|
| | | w/o Format Reward | w/o Diversity Reward | Cross Selection | Majority Voting |
| Qwen3-4b-Base | **78.50** | 78.40 | 78.20 | 76.50 | 77.30 |
| Qwen3-8b-Base | **79.80** | 77.46 | 78.90 | 78.40 | 79.00 |

We conduct ablations to evaluate each component's contribution (Tab. 2). For the Reframer, removing either the diversity rewardor the format reward consistently degraded performance. On Qwen3-8B, removing the diversity reward reduced accuracy from 79.80% to 78.90%, showing that a second view must be meaningfully different to be effective.

For Pseudo-label Selection, we compare our HMS with two simpler strategies: (1) Majority Voting, where branchs uses their majority-voted outputs, and (2) Cross Selection, where branches swap outputs. Both underperform HMS, with Qwen3-8B dropping to 79.00% (Majority) and 78.40% (Cross) vs. 79.80% for HMS.

These trends hold for both 4B and 8B models, confirming the necessity of all core components for *Self-Harmony*'s effectiveness.

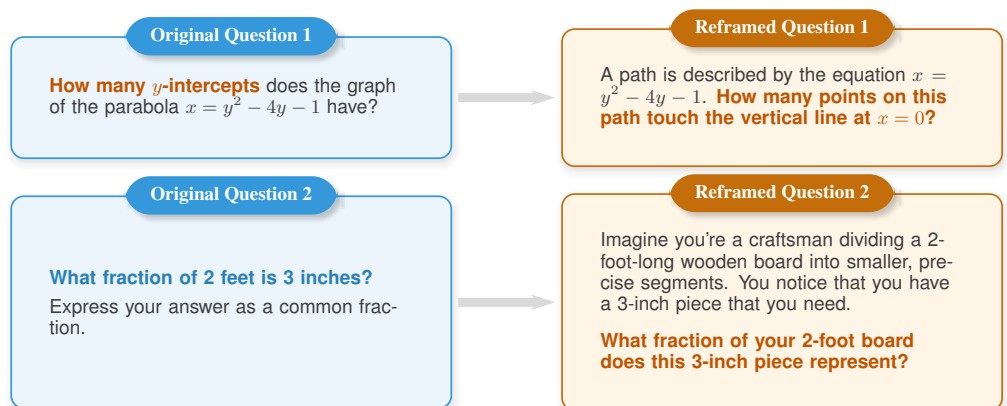

Figure 5: Examples of original and reframed questions. The Reframer rewrites the problem while preserving its semantics and ensuring the correct answer remains unchanged.

### 4.3.3 SENSITIVITY TO REWARD COEFFICIENTS

Our Reframer's reward is governed by two key hyperparameters: the format reward coefficient $w_f$ and the diversity reward coefficient $w_d$. To assess the sensitivity of *Self-Harmony* to these parameters, we conducted a grid search, varying both coefficients from 0.2 to 1.0.

The results, visualized in Figure 4, demonstrate that *Self-Harmony* is robust to the choice of these coefficients. As shown in the heatmap (Figure 4a), the model's performance on Math500 remains high and stable across the entire grid, with accuracy scores consistently above 89.5%. This stability is a significant advantage, as it indicates that our method does not require extensive or precise hyperparameter tuning to achieve strong results. Figure 4b further illustrates this robustness by showing the minimal uncertainty in performance across different settings.

### 4.4 REFRAMED QUESTION EXAMPLES

Figure 5 shows that the Reframer rewrites the problem to introduce stylistic diversity while rigorously preserving the underlying semantic constraints. This confirms the module's ability to perform semantic-invariant transformations, ensuring that the diversity needed for consistency checking is generated without altering the problem's logic. Additional examples are provided in Table 6 and Table 7 in Appendix F.

## 5 CONCLUSION

Effective test-time reinforcement learning (TTRL) is hindered by the challenge of robust pseudo-label selection, as standard methods like majority voting often amplify an LLM's inherent reasoning flaws. In this work, we introduced *Self-Harmony*, a label-free framework built on the core insight that such flaws are often fragile and tied to specific problem phrasings. Through a self-play mechanism, a single model acts as both a Solver and a Reframer to generate a second, paraphrased view of each problem. We show theoretically that the Harmonic Mean Score (HMS) is a robust criterion for identifying the invariant answer across these views, rewarding solutions that are stable under rephrasing. This approach achieves state-of-the-art performance in extensive experiments. Notably, across 30 configurations, *Self-Harmony* ranks first in 28, demonstrating strong generalizability. For example, on MATH500 with Qwen3-1.7B, *Self-Harmony* improves accuracy from 42.7% to 69.6%, while no baseline exceeds 65%, nearly matching the oracle GT-Reward score of 71.8%.

ETHICS STATEMENT

Our work focuses on improving the reasoning capabilities of large language models through a label-free, self-correction mechanism. We primarily use publicly available and widely-cited academic benchmarks (MATH, GSM8K, AIME, AMC, GPQA) and open-source models (Qwen3, Llama-3 series), avoiding the collection of new, sensitive, or personally identifiable data. Our method, Self-Harmony, does not introduce new sources of societal bias; however, it relies on foundational models which may inherit biases from their training data. While our research aims to advance the reliability of AI reasoning, a generally beneficial goal, we acknowledge that any advancement in AI capabilities carries a dual-use risk. We have not identified any direct negative societal impacts or ethical concerns stemming specifically from our proposed method. All research was conducted with a commitment to academic integrity.

REPRODUCIBILITY STATEMENT

To ensure the reproducibility of our results, we have made our implementation publicly available. The complete source code for Self-Harmony(check link in Appendix A), along with scripts to replicate the experiments reported in our paper, is included in the supplementary materials. Our experiments are conducted exclusively on publicly available datasets, including MATH500, GSM8K, AIME, AMC, and GPQA-DIAMOND. The specific open-source models used (e.g., Qwen3-1.7B-Base, Llama-3.1-8B-Instruct) are clearly cited in the main text. Detailed experimental settings, including all hyperparameters, prompt templates, and specific versions of the models, are documented in the appendix to facilitate replication. Our evaluation metric, pass@1, is standard for these benchmarks, and our comparison with baseline methods uses identical learning hyperparameters to ensure fairness.

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

## A    CODE

The code is publicly available at `https://github.com/physicsru/self_harmony`. Follow the instructions to run our code and reproduce our results.

## B    VISUALIZATION

### B.1    PSEUDO LABEL SELECTION

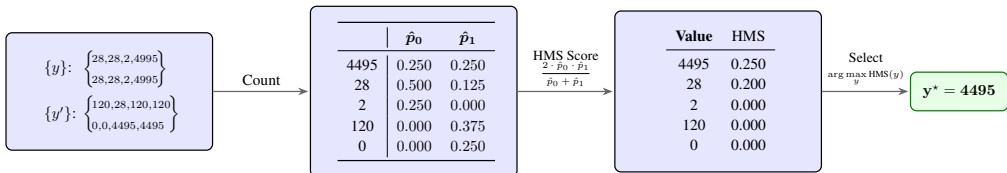

Figure 6: Pseudo label selection with pre-computed harmonic mean values.

The psuedo label selection is like following and shown in Figure 6, counting, calculating the frequency, calculating the harmonic mean score, picking the candidate with largest harmonic mean as pseudo label.

### B.2    FUNCTION OF DIVERSITY REWARD

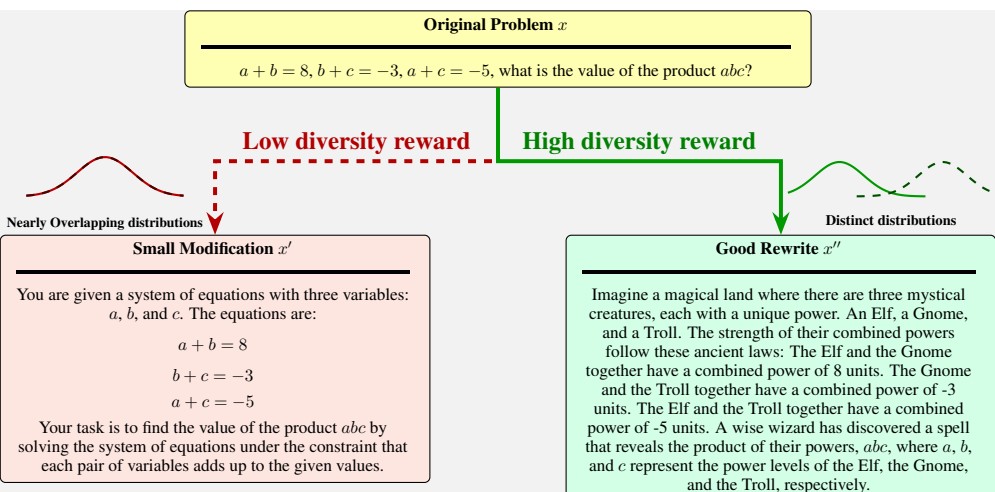

Figure 7: The impact of the diversity reward on the Reframer policy. Without a strong diversity signal, the model may produce trivial modifications ($x'$) that fail to test robustness. The diversity reward, based on the Jensen-Shannon Divergence between answer distributions, incentivizes the model to generate creative and semantically rich reformulations ($x''$) that provide a more challenging and effective consistency check.

The diversity reward incentivize the reframer to reframe a question in a non-trivial way with distribution distance, shown in Figure 7

## C   PSEUDO LABEL CROSS DIFFERENT DIFFICULTIES

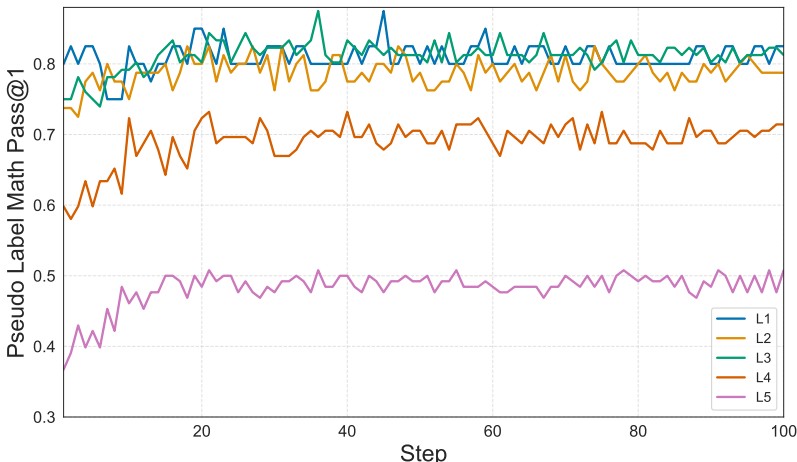

Figure 8: Pseudo-label accuracy generated by our Harmonic Mean Score over 100 training steps, categorized by math problem difficulty (L1 = easiest, L5 = hardest). The accuracy shows a clear correlation with problem complexity, achieving high performance on levels L1-L3.

## D   IMPLEMENTATION DETAILS

### D.1   PROMPT DESIGN

We design two distinct prompts for the **Solver** and the **Reframer** components.

**Solver.**   During both training and evaluation, the Solver is guided by a straightforward instruction-following prompt to promote clear, step-by-step reasoning:

```
Let's think step by step and output the final answer within \boxed{}.
```

**Reframer.**   The Reframer is tasked with creatively rephrasing the original problem before solving it, promoting diversity and robustness. The prompt explicitly instructs the model to first transform the problem using one of several strategies, then solve the transformed version. The transformation must preserve logical and numerical consistency with the original problem, while discouraging trivial paraphrasing. The full prompt is shown below:

```
You are The Reframer, an AI that solves problems with creative flair.
Your challenge is not just to find the answer, but to find a more
interesting way to get there.

Mission:
1. Transform: DO NOT solve the ORIGINAL PROBLEM directly.
   Rewrite it using one of the strategies below to unlock a
   new perspective.
2. Solve: Solve the new problem you created. Your solution's
   style must match your chosen transformation.

Strategies (Toolkit):
- Concretize: Turn an abstract idea into a tangible story.
- Generalize: Convert a specific case into a general formula.
- Domain Shift: Change the domain (e.g., math -> code,
  logic -> game).
- Add Noise: Introduce red herrings to test focus.
- Reverse: Start from the final result and work backward.
- Incremental Complexity: Solve a simpler warm-up problem
```

```
 first, then apply the pattern to the original numbers.
- Focus on Constraints: Frame as a puzzle with strict rules.
- POV Shift: Narrate from the perspective of an element
  inside the problem.

Rules:
- State your chosen Strategy first.
- Core logic, numbers, and the final boxed answer must match
  the ORIGINAL PROBLEM.
- Avoid copying 5+ consecutive words from the original text.

Response Format:
STRATEGY: [Chosen strategy]

REWRITTEN PROBLEM:
[Your transformed problem statement.]

SOLUTION:
[Step-by-step solution matching the transformed problem.]
\boxed{[Final answer]}

ORIGINAL PROBLEM:
```

### D.2 Experiment Details

First, for all baselines, we use the official code provided in their public repositories. For Co-Reward, we use models of the same size to rewrite questions, instead of relying on Qwen3-32B, to ensure a fair comparison. When training the Qwen3-XB-Base series, we use the corresponding instruct version of Qwen3-XB for question rewriting. Similarly, for the Llama-Instruct series, we use their respective instruct models. The prompt used for question rewriting follows the one provided in the original paper Zhang et al. (2025).

**Hyperparameters** he following hyperparameters are shared across all baseline methods and Self-Harmony. For the format coefficient and diversity coefficient, ususally we set as 0.1 for most cases. And few of experiment using other setting, check 4a for setting these two parameters.

## E    Extra Experiments

There are some extra studies in this section.

### E.1    Convergence of Pseudo-Label Selectors

At each training step, we use the identical set of rollouts from both views to determine the pseudo-label chosen by two different methods: dual-view majority voting and our proposed Harmonic Mean Score (HMS) selector. Figure 9 plots the difference in Pass@1 accuracy between these two selected labels.

The plot shows that HMS initially selects more accurate labels, providing a crucial advantage when the model is uncertain. The difference between the two methods converges to zero because our overall training dynamic successfully stabilizes the model's policy. As the model becomes more confident and its answer distribution sharpens, both majority voting and HMS will naturally select the same high-probability answer, causing their choices to converge. This convergence is evidence of a successfully stabilized policy, initiated by the early robustness of the HMS criterion.

### E.2    Validating Semantic Consistency of the Reframer

A critical requirement for Self-Harmony is that the Reframer generates problems that are semantically equivalent to the originals. To validate this, we conducted an analysis using powerful, external Large Language Models (LLMs) as unbiased judges. We randomly sampled 200 problems and used

Table 3: Hyperparameters.

| Hyperparameter | Value |
|---|---|
| Max prompt length | 1024 |
| Max response length | 3072 |
| Batch size | 128 GSM8K
128 MATH500
80 AMC
24 AIME
32 MMLU-Pro
192 GPQA |
| Policy mini batch size | 128 GSM8K
128 MATH500
80 AMC
24 AIME
32 MMLU-Pro
192 GPQA |
| Policy micro batch size per GPU | 8 GSM8K
8 MATH500
5 AMC
3 AIME
4 MMLU-Pro
4 GPQA |
| Learning rate | $3 \times 10^{-6}$ |
| LR warmup steps ratio | 0.1 |
| hline Weight decay | 0.01 |
| Learning rate warmup | cosine |
| Optimizer | Adam |
| Temperature | 1.0 for Training
0.8 for Testing |
| Top $k$ | -1 |
| Top $p$ | 1 Training
0.95 Testing |
| Number of samples per example $n$ | 16 GSM8K
16 MATH500
32 AMC
32 AIME
16 MMLU-Pro
16 GPQA |
| Number of samples per example for Testing | 16 |
| Remove padding | True |
| Use KL loss | True |
| KL loss coefficient | 0.005 |
| KL loss type | low var kl |
| Clip ratio | 0.2 |
| Grad clip | 1.0 |
| Temperature | 1.0 Training
0.8 Testing |
| Verifier | MATH500 verl math verifier
GSM8K verl math verifier
AIME ttrl math verifier
AMC ttrl math verifier
GPQA ttrl math verifier
MMLU-Pro ttrl math verifier |

our trained Reframer to generate their rewritten counterparts. We then prompted two state-of-the-art models, GPT-4o and Claude 4 Sonnet, to solve both the original and the rewritten versions.

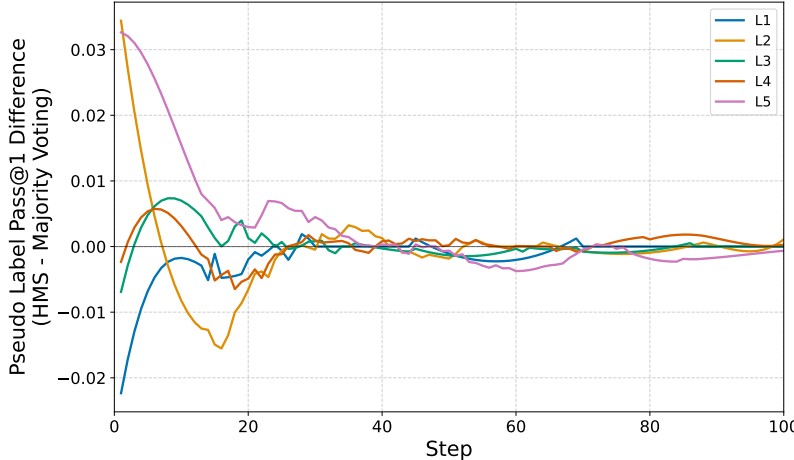

Figure 9: Pass@1 accuracy difference between labels selected by HMS and dual-view majority voting, using the exact same rollouts at each step. The convergence towards zero indicates that as the model's policy stabilizes, both selection methods increasingly agree.

Table 4: Analysis of Reframer-generated problems using external oracle models(GSM8K). The high consistency rates confirm that the rewritten problems are semantically equivalent to the originals.

| Metric (Pass@1 %) | GPT-4o | Claude 4 Sonnet |
|---|---|---|
| Original Questions | 98.41 | 99.24 |
| Rewframed Questions | 87.30 | 87.88 |
| **Consistency Rate** | **88.71** | **88.55** |

Our primary metric is the **Consistency Rate**, which measures the percentage of cases where the external oracle model produced the same final answer for both the original and rewritten versions of a problem. The results, presented in Table 4, show a high degree of consistency. For GPT-4o, the consistency rate was **88.71%**, and for Claude 4 Sonnet, it was **88.55%**.

This strong agreement demonstrates that our Reframer reliably generates new problems that preserve the core semantic meaning of the originals. This validation is crucial, as it confirms that the second view used in our Harmonic Mean Score calculation is a valid and meaningful perspective on the original problem.

### E.3 SENSITIVITY ANALYSIS ON SEMANTIC SHIFT

Smaller language models (0.6B) are more likely to introduce semantic drift, wording biases, or shallow perturbations when paraphrasing, compared to larger models (14B). To test whether such imperfections could cause the Harmonic Mean selector to incorrectly down-weight valid answers, we conduct a stress test on the Confident-Correct subset of the math dataset. This subset is constructed by running Qwen3 models with 16-rollout majority voting and keeping only questions where the majority count exceeds half the rollouts and the majority answer is correct.

For each Solver (Qwen3-1.7B-Base, Qwen3-4B-Base, Qwen3-8B-Base), we fix the Solver and vary the Reframer size across 0.6B, 1.7B, 4B and 14B parameters. This uses model scale as a proxy for paraphrase quality: the 0.6B Reframer tends to produce lower-fidelity, noisier paraphrases, while the 14B Reframer yields higher semantic fidelity. If Self-Harmony were highly sensitive to semantic shift, we would expect performance to degrade when using small Reframers on this already-correct subset.

Instead, we observe consistent and substantial gains over the baseline across all Solvers and all Reframer sizes (Table 5). Even with the 0.6B Reframer, Self-Harmony significantly improves Mean Pass@1 on the Confident-Correct subset (like Qwen3-4B-Base: 83.98% → 96.48%; Qwen3-8B-

Table 5: Self-Harmony performance on the Confident-Correct subset under different Reframer scales. For each Solver, we fix the Solver and vary the Reframer size; Baseline denotes initial model.

| Self-Harmony / Confident-Correct | Baseline | 0.6B | 1.7B | 4B | 14B |
|---|---|---|---|---|---|
| Qwen3-1.7B-Base | 86.32 | 94.92 | 94.92 | 95.31 | 96.48 |
| Qwen3-4B-Base | 83.98 | 96.48 | 97.65 | 98.04 | 98.36 |
| Qwen3-8B-Base | 89.06 | 97.85 | 97.75 | 98.02 | 99.07 |

Base: 89.06% → 97.85%). This demonstrates that the Harmonic Mean selector is **not overly sensitive** to small perturbations or imperfect semantic equivalence: it continues to recognize and preserve valid answers despite noisy rephrasings. Performance further improves as Reframer size increases (up to 99.07% with Qwen3-8B-Base + 14B Reframer), indicating that even the smallest 0.6B Reframer is already sufficient to support Self-Harmony, and larger Reframers bring additional gains.

### E.4 Harmonic Mean Pseudo Label Selector vs. Harmonic Mean Reward

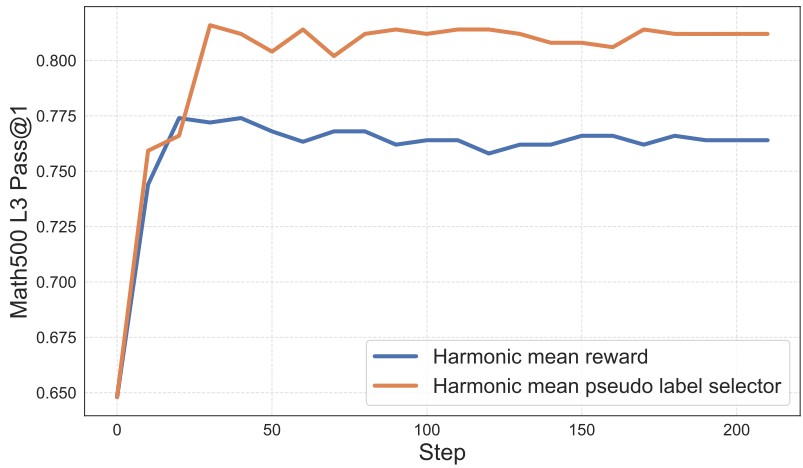

Figure 10: Performance comparison on the Math500 Pass@1 metric between the harmonic mean pseudo label selector and the harmonic mean reward baseline over training steps.

Since there are some paper only using internal signal rather than choosing a pseudo label, an ablation study is conducted to compare the performance for harmonic mean in two different using way: 1. Pseudo label selector 2. Harmonic mean as reward. Experiment shows that pseudo label way has better performance as shown in Figure 10

### E.5 Additive Reward vs. Multiplicative Reward

To further investigate the effect of different reward aggregation strategies, we compare the performance of the *additive reward* and *multiplicative reward* schemes. The additive reward uses a weighted sum of individual loss components, with weights set to match the proportional influence of each component in the multiplicative reward. Specifically, we aligned the weights based on the effective gradients derived from a first-order Taylor expansion. Our multiplicative reward is defined as:

$$R_{\text{mult}} = (1 - w_f R_f)(1 - w_d R_d) R_{\text{solve}}$$

For correct answers ($R_{\text{solve}} = 1$) and small penalty terms, this linearizes to:

$$R_{\text{mult}} \approx R_{\text{solve}} - w_f R_f - w_d R_d$$

To control for hyperparameter influence, we configured the additive baseline ($R_{\text{add}} = w_3 R_{\text{solve}} + w_1 R_f + w_2 R_d$) to match this approximation strictly: $w_3 = 1 w_1 = -w_f w_2 = -w_d$. This align-

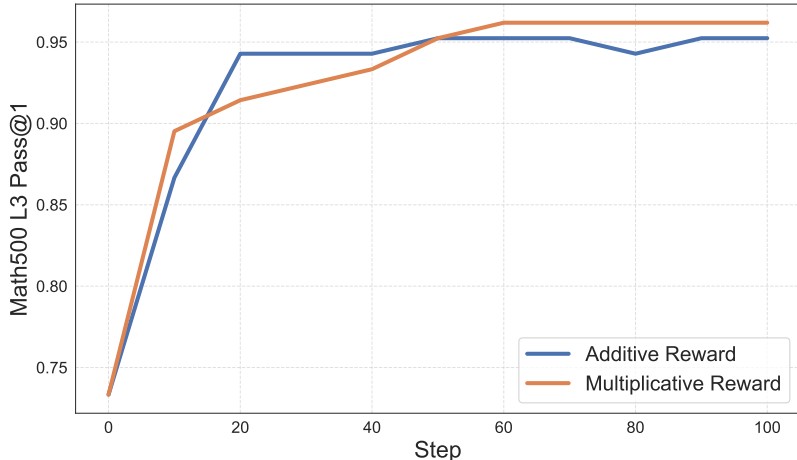

Figure 11: Comparison between the **additive reward** (blue) and **multiplicative reward** (orange) formulations on the Math500 L3 Pass@1 metric. The additive reward combines components through a weighted sum, while the multiplicative reward uses the product of the same components ratio. Both rewards are normalized to 1 for fair comparison, and all other training parameters are kept exactly the same.

ment ensures that both models are subject to identical penalty magnitudes for format and diversity violations. Consequently, the superior performance of Self-Harmony can be attributed to the structural advantage of the multiplicative design (which acts as a strict "AND" gate) rather than disparate hyperparameter settings. For a fair and controlled comparison, both rewards are normalized to 1, and all other hyperparameters and training settings are kept exactly the same. As shown in Figure 11, while both methods achieve similar final performance, the multiplicative reward demonstrates slightly faster convergence in the early stages of training.

## F  EXAMPLES OF REWRITTEN PROBLEMS

To illustrate the role of the *Reframer*, we present examples of original and rewritten problems. Well-formed rewrites are shown in Table 6, while problematic cases are shown in Table 7. In the well-formed examples, the rewritten questions modify the narrative or context but remain functionally equivalent to the originals, ensuring that the correct answers are preserved. In contrast, the problematic rewrites fail to meet these requirements, either by reducing to trivial restatements or by altering the correct answers, as demonstrated in Table 7.

## G  THEORETICAL DISCUSSION

### G.1  PROOF OF THEOREM 3.2

**Theorem G.1** (Harmonic Mean Selector from Invariant Infomax). *Assume the* view-invariance *condition (Assumption 3.1) holds. Suppose moreover that the following conditions are satisfied.*

*Non-degeneracy. For every label $a$,*

$$p_0(a) + p_1(a) < 1 \tag{1}$$

*This simply excludes the trivial case in where the majority voting can solve.*

*Balanced–Confidence. There exists a constant $\kappa \in (0, 1)$ such that for every maximiser $a^\star$ of $J_\lambda(\cdot)$*

$$\left| p_0(a^\star) - p_1(a^\star) \right| \leq \kappa \left[ p_0(a^\star) + p_1(a^\star) \right]. \tag{2}$$

*Then, for the penalty weight $\lambda = 2$, the pseudo label that maximises the second-order approximation of the* view-invariant Infomax *objective $J_2(a) = I(Z_a; A) - 2\, I(Z_a; X)$ is obtained by the harmonic*

| Original Question | Rewritten Question |
|---|---|
| Let $z$ be a complex number such that $|z| = 1$. Find the maximum value of $$|1 + z| + |1 - z + z^2|.$$ | Imagine you're an elf tasked with finding the maximum healing power of two magical spells combined. One spell's strength is represented by $|1 + z|$, where $z$ is a magical rune with a known power level of 1 ($|z| = 1$). The other spell's strength is $|1 - z + z^2|$. Your job is to figure out the maximum total healing power when using both spells together. |
| How many $y$-intercepts does the graph of the parabola $x = y^2 - 4y - 1$ have? | A path is described by the equation $x = y^2 - 4y - 1$. How many points on this path touch the vertical line at $x = 0$? |
| A train has 172 people traveling on it. At the first stop 47 people get off and 13 more people get on, and at the next stop another 38 people get off. How many people are on the train? | Imagine a train journey where the train starts with 172 people. The train first stops at a station where 47 passengers disembark and 13 new passengers board the train. At the next station, 38 more passengers get off. How many people remain on the train as it continues its journey? |
| A whirligig spins at five times the speed of a thingamabob. A whatchamacallit spins eleven times faster than a thingamabob. A whatchamacallit spins at 121 meters per second. How fast does a whirligig spin? | Imagine the relationship between these spinning objects as a programming task. The thingamabob is like a base class ('BaseObject') from which other objects inherit properties. A whirligig spins at five times the speed of this base class, making it a derived class ('FiveTimesObject'). Similarly, a whatchamacallit spins eleven times faster ('ElevenTimesObject'), inheriting the speed from 'BaseObject'. Given that the 'ElevenTimesObject' spins at 121 meters per second, determine the speed of the 'FiveTimesObject'. |
| What fraction of 2 feet is 3 inches? Express your answer as a common fraction. | Imagine you're a craftsman dividing a 2-foot-long wooden board into smaller, precise segments. You notice that you have a 3-inch piece that you need. What fraction of your 2-foot board does this 3-inch piece represent? |
| What fraction of 2 feet is 3 inches? Express your answer as a common fraction. | Imagine you are designing a digital game where objects need to be connected at specific distances. In the game, the main character moves by taking steps of 2 feet, but the tasks require the character to interact with objects placed every 3 inches. What fraction of the main character's step length (2 feet) should be used to reach the nearest interaction point (3 inches)? Express your answer as a common fraction. |
| Let $N$ be the greatest four-digit positive integer with the property that whenever one of its digits is changed to 1, the resulting number is divisible by 7. Let $Q$ and $R$ be the quotient and remainder, respectively, when $N$ is divided by 1000. Find $Q + R$. | Find the greatest four-digit positive integer $N$ such that changing any one of its digits to 1 results in a number that is divisible by 7. Then, determine the quotient $Q$ and remainder $R$ when $N$ is divided by 1000, and compute $Q + R$. |
| Find the value of $r$ such that $$\frac{6r^2 - 19r - 7}{2r - 7} = 4r - 3.$$ | Find a general formula to solve the equation $\frac{6r^2 - 19r - 7}{2r - 7} = 4r - 3$ for $r$, and then apply it to find the specific value of $r$. |
| Each vertex of a regular octagon is independently colored either red or blue with equal probability. The probability that the octagon can then be rotated so that all of the blue vertices end up at positions where there were originally red vertices is $\frac{m}{n}$, where $m$ and $n$ are relatively prime positive integers. What is $m + n$? | Consider a regular octagon with vertices labeled as $0, 1, 2, ..., 7$. We color each vertex either red or blue with equal probability (which we can think of as a coin flip). After painting, we can rotate the octagon (which we can think of as performing arithmetic modulo 8) to potentially reposition the blue vertices onto positions originally occupied by red vertices. What is the probability of this happening, expressed in the form $\frac{m}{n}$ where $m$ and $n$ are coprime integers, and find $m + n$? |
| Find the coordinates of point $P$ on the line $y = -x + 6$ such that the distances from $P$ to points $A(10, -10)$ and $O(0, 0)$ are equal, i.e., $PA = PO$. Express your answer as an ordered pair $(a, b)$. | Consider a 2D plane where the line y = -x + 6 is defined. Let's shift the domain to the set of all complex numbers such that the imaginary part corresponds to the y-coordinate and the real part corresponds to the x-coordinate. We need to find a complex number $P = a + bi$ on the line $y = -x + 6$ (which becomes $b = -a + 6$) that is equidistant from the complex numbers representing points $A(10, -10)$ and $O(0, 0)$. This means finding $P$ such that the distances $|P - A|$ and $|P - O|$ are equal. Express $P$ as $(a, b)$ in the original plane coordinates. |

Table 6: Examples of original and well-formed rewritten problem statements. The rewrites are functionally equivalent to the originals, ensuring that the correct answer remains unchanged.

| Original Question | Rewritten Question |
|---|---|
| Find the coordinates of point $P$ on the line $y = -x + 6$ such that the distances from $P$ to points $A(10, -10)$ and $O(0,0)$ are equal, i.e., $PA = PO$. Express your answer as an ordered pair $(a, b)$. | Find the coordinates of point $P$ on the line $y = -x + 6$ such that the distances from $P$ to points $A(10, -10)$ and $O(0,0)$ are equal, i.e., $PA = PO$. Express your answer as an ordered pair $(a, b)$. |
| Based on these matrices, choose the correct statement. Options: A. Z and X represent observables. B. $(e^X) * Y * (e^{-X})$ represents a quantum state. C. W and X represent the evolution operator of some quantum system. D. There exists a vector to which if one multiplies $e^X$, the norm of the vector changes. | Imagine you're watching a dramatic play called Q-Matrix. The different acts in this play are represented by matrices W, X, Y, and Z. The director Z calls out, "W and X, you two are my favorite scenes. Any other order, pfft, they're iust smoke and mirror tricks!" What does the director mean by this? |
| Let $a, b, c, d$ be positive real numbers. Find the minimum value of $$(a + b + c + d)\left(\frac{1}{a} + \frac{1}{b} + \frac{1}{c} + \frac{1}{d}\right).$$ | Let $S$ be a set of positive real numbers. Find the minimum value of $S\left(\frac{1}{S}\right)$ where $S$ is a sum of $n$ positive real numbers. Assume that $S \leq n^2$. |
| Let $a, b, c, d$ be positive real numbers. Find the minimum value of $$(a + b + c + d)\left(\frac{1}{a} + \frac{1}{b} + \frac{1}{c} + \frac{1}{d}\right).$$ | Suppose that four kids carry water from the kitchen to their bath tubs in their houses. The four containers have capacities a, b, c and d litres, respectively. The kids have to carry enough water to fill their bath per a + b + c + d litres of water filled and each kid takes a turn to carry the water. When he carries water, he uses 1/a, 1/b, 1/c and 1/d of time to carry water per bath which is equal to the number of turns each gone the bath tub respectively. What is the minimum total number of turns which sum up to fill a + b + c + d of water when each kid uses their own specific container? Moreover, during each time unit, the whole amount of water a, b, c and d liters is filled and drained from the containers, such that the water level in each container remains the same. How many minutes does it take until every kid has filled their bath tub? |

Table 7: Examples of original and problematic rewritten problem statements. These rewrites either reduce to trivial restatements or alter the correct answer, violating functional equivalence.

*mean of the two view-probabilities:*

$$y^\star = \arg\max_a J_2(a) = \arg\max_a \frac{2 \, p_0(a) \, p_1(a)}{p_0(a) + p_1(a)}.$$

*Proof of Theorem 3.2.* First of all, let's define the two view and objective function:

The two views are defined by the random variable $X \in \{x, x'\}$, sampled uniformly, so that

$$p(X = x) = p(X = x') = \tfrac{1}{2}.$$

For a candidate answer $a$, write

$$p_0(a) = p(A = a \mid X = x), \qquad p_1(a) = p(A = a \mid X = x').$$

We shall also use

$$\bar{p}(a) = \tfrac{1}{2}\big(p_0(a) + p_1(a)\big), \qquad \delta(a) = \tfrac{1}{2}\big(p_0(a) - p_1(a)\big),$$

and the Bernoulli indicator $Z_a = \mathbb{I}\{A = a\}$.

Based on infomax principal, we introduce the invariant-Infomax score to be maximised:

$$J_\lambda(a) = I(Z_a; A) - \lambda \, I(Z_a; X) \tag{3}$$

and the Theorem 3.2 sets $\lambda = 2$, where the objective function is

$$J_2(a) = I(Z_a; A) - 2 \, I(Z_a; X) \tag{4}$$

And now let's expand mutual information expression:

with mutual information defined in two equivalent ways

$$I(U;W) = H(U) - H(U \mid W) = \mathbb{E}_{w \sim W}\big[ D_{\mathrm{KL}}\big(p(U \mid W = w) \,\|\, p(U)\big)\big]. \qquad (\star)$$

Taking $U = Z_a$, $W = A$ in $(\star)$:

$$I(Z_a; A) = H(Z_a) - H(Z_a \mid A).$$

Because $Z_a$ is a deterministic function of $A$ (knowing $A$ tells us with certainty whether $A = a$), the conditional entropy vanishes: $H(Z_a \mid A) = 0$. Consequently

$$I(Z_a; A) = H(Z_a).$$

Marginally $Z_a$ is Bernoulli with mean $p(Z_a = 1) = p(A = a) = \bar{p}(a)$, so

$$H(Z_a) = h\big(\bar{p}(a)\big), \quad \text{where} \quad h(x) = -x \ln x - (1 - x) \ln(1 - x)$$

is the binary entropy.

Using the second form in $(\star)$ with $U = Z_a$, $W = X$ we obtain

$$I(Z_a; X) = \sum_X p(X)\, D_{\mathrm{KL}}\big(p(Z_a \mid X) \,\|\, p(Z_a)\big). \qquad (5)$$

– *Conditional laws.* For $X = x$ the answer distribution is $p_0(a)$, For $X = x'$ the answer distribution is $p_1(a)$; therefore $Z_a \mid X = x \sim \mathrm{Bern}(p_0(a))$. Similarly $Z_a \mid X = x' \sim \mathrm{Bern}(p_1(a))$.

– *Marginal law.* By total probability and the uniform prior on $X$, $p(Z_a = 1) = \frac{1}{2}p_0(a) + \frac{1}{2}p_1(a) = \bar{p}(a)$, so $Z_a \sim \mathrm{Bern}(\bar{p}(a))$.

– *Substituting in equation 5.* Denoting $\mathrm{Bern}(p)$ a Bernoulli distribution of mean $p$,

$$I(Z_a; X) = \tfrac{1}{2} D_{\mathrm{KL}}\big(\mathrm{Bern}(p_0(a)) \,\|\, \mathrm{Bern}(\bar{p}(a))\big) + \tfrac{1}{2} D_{\mathrm{KL}}\big(\mathrm{Bern}(p_1(a)) \,\|\, \mathrm{Bern}(\bar{p}(a))\big).$$

Let $q = \bar{p}(a)$ for brevity and define

$$g(p) = D_{\mathrm{KL}}\big(\mathrm{Bern}(p) \,\|\, \mathrm{Bern}(q)\big) = p \ln \frac{p}{q} + (1 - p) \ln \frac{1 - p}{1 - q}.$$

Taylor-expand $g$ around $p = q$:

$$g'(p) = \ln \frac{p}{q} - \ln \frac{1 - p}{1 - q}, \qquad g''(p) = \frac{1}{p} + \frac{1}{1 - p},$$

so that $g(q) = g'(q) = 0$ and $g''(q) = 1/[q(1 - q)]$. For $\Delta = p - q$ small,

$$g(q + \Delta) = \frac{\Delta^2}{2q(1 - q)} + \mathcal{O}(\Delta^3).$$

Since $p_0 = q + \delta$ and $p_1 = q - \delta$ we have $g(p_0) = g(p_1) = \delta^2/[2q(1 - q)] + \mathcal{O}(\delta^3)$. Averaging the two contributions yields

$$I(Z_a; X) = \frac{\delta(a)^2}{2\,\bar{p}(a)\big(1 - \bar{p}(a)\big)} + \mathcal{O}\big(\delta(a)^3\big). \qquad (6)$$

So Second-order Taylor expansion approximation is:

$$J_2(a) = I(Z_a; A) - 2\,I(Z_a; X) \approx h(\hat{p}(a)) - \frac{\delta(a)^2}{\bar{p}(a)\big(1 - \bar{p}(a)\big)}. \qquad (7)$$

Using Assumption 2, we expand the denominator $\frac{1}{1-\bar{p}(a)} = 1 + \bar{p}(a) + \bar{p}(a)^2 + \cdots$ and obtain:

$$\frac{\delta(a)^2}{\bar{p}(a)\big(1-\bar{p}(a)\big)} = \frac{\delta(a)^2}{\bar{p}(a)} - \underbrace{\delta(a)^2\big(\bar{p}(a) + \bar{p}(a)^2 + \cdots\big)}_{R(a)}.$$

Balanced confidence (Assumption 2) implies $|\delta(a)| \leq \kappa\,\bar{p}(a)$, so

$$|R(a)| \leq \frac{\kappa^2\bar{p}(a)^2}{1-\bar{p}(a)} \leq 2\kappa^2\bar{p}(a)^2 = O\big(\bar{p}(a)^2\big),$$

which is third-order. Discarding $R(a)$ gives the second-order approximation

$$J_2(a) \approx h\big(\bar{p}(a)\big) - \frac{\delta(a)^2}{\bar{p}(a)}.$$

Next, by Assumption 1, we have $\bar{p}(a) < \frac{1}{2}$ for all $a$, and since the binary entropy $h$ is strictly increasing on $(0, \frac{1}{2})$, maximising $h(\bar{p}(a))$ is equivalent to maximising $\bar{p}(a)$ itself. Therefore

$$\arg\max_a J_2(a) = \arg\max_a \left[\bar{p}(a) - \frac{\delta(a)^2}{\bar{p}(a)}\right].$$

Finally, using the identity

$$\bar{p}(a) - \frac{\delta(a)^2}{\bar{p}(a)} = \frac{2\,p_0(a)\,p_1(a)}{p_0(a) + p_1(a)},$$

we define the harmonic–mean surrogate score:

$$S(a) = \frac{2\,p_0(a)\,p_1(a)}{p_0(a) + p_1(a)}.$$

Thus, the pseudo-label selected by $J_2$ satisfies

$$y^\star = \arg\max_a \frac{2\,p_0(a)\,p_1(a)}{p_0(a) + p_1(a)} \in \arg\max_a J_2(a) \tag{8}$$

The proof only assumes two conditional answer distributions, $p_0(a) = p(a \mid X = x)$ and $p_1(a) = p(a \mid X = x')$, indexed by a binary view variable $X$. These views can be instantiated in multiple ways:

(i) two semantically equivalent queries, $p_0(a) = p(a \mid x)$ and $p_1(a) = p(a \mid x')$;

(ii) the same query under different system prompts $\sigma$, $\sigma'$, $p_0(a) = p(a \mid x, \sigma)$ and $p_1(a) = p(a \mid x, \sigma')$;

or any other mechanism that induces two correlated yet distinct answer distributions. Because the derivation treats $X$ abstractly, the harmonic-mean selector remains valid for all such pairings.

$\square$

## G.2    GENERALIZATION OF THEOREM 1: TUNABLE $\lambda$ AND GENERALIZED MEANS

In this part, we extend the derivation of the Invariant Infomax objective to the general case where the penalty weight $\lambda$ is tunable. We demonstrate that varying $\lambda$ is theoretically equivalent to selecting a specific **Generalized Mean** for aggregating binary view probabilities. Recall the Taylor expansion of the Invariant Infomax objective $J_\lambda(a) = I(Z_a; A) - \lambda I(Z_a; X)$. Using the second-order approximations derived in the main proof, the objective simplifies to:

$$J_\lambda(a) \approx \bar{p}(a) - \lambda\frac{\delta(a)^2}{2\bar{p}(a)}, \tag{9}$$

where $\bar{p}(a) = \frac{p_0(a)+p_1(a)}{2}$ is the mean probability and $\delta(a) = \frac{p_0(a)-p_1(a)}{2}$ is the semi-difference between views. Note that $\delta(a)^2$ is exactly equivalent to the statistical variance of the two view probabilities.

Now, consider the **Generalized Mean** with exponent $k$, defined as $M_k(p_0, p_1) = \left( \frac{p_0^k + p_1^k}{2} \right)^{1/k}$. The second-order Taylor expansion of $M_k$ around the mean $\bar{p}$ is given by:

$$M_k(p_0, p_1) \approx \bar{p}(a) + (k-1)\frac{\delta(a)^2}{2\bar{p}(a)}. \tag{10}$$

By equating the coefficients of the variance term $\delta(a)^2$ in Eq. equation 9 and Eq. equation 10, we establish a direct mapping between the penalty weight $\lambda$ and the mean exponent $k$:

$$-\lambda = k - 1 \implies k = 1 - \lambda. \tag{11}$$

This mapping allows us to categorize the behavior of the selector across the spectrum of $\lambda$. In the list below, we describe the penalty regarding the variance term $\delta(a)^2$:

- $\lambda = -1 \implies k = 2$ **(Quadratic Mean / RMS):** The objective effectively *adds* a variance bonus ($\approx \bar{p} + \frac{\delta^2}{2\bar{p}}$). This is "risk-seeking" and prefers disagreement, making it unsuitable for consistency checking.

- $\lambda = 0 \implies k = 1$ **(Arithmetic Mean):** The objective maximizes $\bar{p}$ (Standard Voting). It ignores view consistency entirely (coefficient of $\delta^2$ is 0).

- $\lambda = 1 \implies k = 0$ **(Geometric Mean):** The objective maximizes $\sqrt{p_0 p_1}$. It penalizes variance with a coefficient of $-1$ ($\approx \bar{p} - \frac{\delta^2}{2\bar{p}}$).

- $\lambda = 2 \implies k = -1$ **(Harmonic Mean):** The objective maximizes $\frac{2p_0 p_1}{p_0 + p_1}$. It penalizes variance with a coefficient of $-2$ ($\approx \bar{p} - 2\frac{\delta^2}{2\bar{p}}$), effectively doubling the penalty compared to the Geometric Mean.

### G.3 Multiple Views Case and Generalized Harmonic Mean

In this part, we extend the analysis of the Harmonic Mean Selector from the binary view setting ($K = 2$) to the general multi-view setting ($K \geq 2$). We show that maximizing the view-invariant Infomax objective with $\lambda = 2$ is theoretically equivalent to maximizing the *Generalized Harmonic Mean* of the view probabilities.

**Theorem G.2** (Generalized Harmonic Mean Selector). *Let there be $K$ views defined by a random variable $X$ uniformly distributed over $\{x_1, \ldots, x_K\}$. Let $p_k(a) = p(A = a \mid X = x_k)$ be the prediction probability for candidate $a$ under view $k$. Under the same assumptions of non-degeneracy and balanced confidence as Theorem 3.2, maximizing the second-order approximation of the objective $J_2(a) = I(Z_a; A) - 2I(Z_a; X)$ recovers the Generalized Harmonic Mean of the $K$ probabilities:*

$$y^\star = \arg \max_a \left( \frac{1}{K} \sum_{k=1}^{K} \frac{1}{p_k(a)} \right)^{-1}.$$

*Proof.* We analyze the objective function and the harmonic mean separately using Taylor expansions around the arithmetic mean. Let the arithmetic mean be $\bar{p}(a) = \frac{1}{K} \sum_{k=1}^{K} p_k(a)$ and the variance across views be $\sigma^2(a) = \frac{1}{K} \sum_{k=1}^{K} (p_k(a) - \bar{p}(a))^2$.

1. Expansion of the Infomax Objective. The mutual information $I(Z_a; A)$ remains the entropy of the marginal $Z_a$, which approximates to the mean probability for small $p$:

$$I(Z_a; A) = H(\bar{p}(a)) \approx \bar{p}(a) \quad \text{(up to first order)}.$$

The mutual information between the pseudo-label and the views, $I(Z_a; X)$, is the average KL-divergence between the specific view distributions and the mean distribution:

$$I(Z_a; X) = \frac{1}{K} \sum_{k=1}^{K} D_{\mathrm{KL}}\big(\mathrm{Bern}(p_k(a)) \,\|\, \mathrm{Bern}(\bar{p}(a))\big).$$

Using the second-order Taylor expansion of the KL-divergence, $D_{\mathrm{KL}}(p \parallel q) \approx \frac{(p-q)^2}{2q(1-q)}$, we obtain:

$$I(Z_a; X) \approx \frac{1}{K} \sum_{k=1}^{K} \frac{(p_k(a) - \bar{p}(a))^2}{2\bar{p}(a)(1 - \bar{p}(a))} = \frac{\sigma^2(a)}{2\bar{p}(a)(1 - \bar{p}(a))}.$$

Applying the simplification from the proof of Theorem 3.2 (where the denominator term $1 - \bar{p}(a)$ is dominated by the first order behavior), the objective with penalty $\lambda = 2$ becomes:

$$J_2(a) = I(Z_a; A) - 2I(Z_a; X) \approx \bar{p}(a) - \frac{\sigma^2(a)}{\bar{p}(a)}. \tag{12}$$

2. Expansion of the Generalized Harmonic Mean. Let $H_K(a)$ denote the harmonic mean of probabilities $\{p_1(a), \ldots, p_K(a)\}$. We approximate terms of the form $1/p_k(a)$ using a Taylor expansion of $f(x) = 1/x$ around the mean $\bar{p}(a)$:

$$\frac{1}{p_k} \approx \frac{1}{\bar{p}} - \frac{(p_k - \bar{p})}{\bar{p}^2} + \frac{(p_k - \bar{p})^2}{\bar{p}^3}.$$

Averaging these inverse terms over $K$ views:

$$\frac{1}{K} \sum_{k=1}^{K} \frac{1}{p_k} \approx \frac{1}{\bar{p}} - \underbrace{\frac{1}{K\bar{p}^2} \sum (p_k - \bar{p})}_{0} + \frac{\sigma^2}{\bar{p}^3} = \frac{1}{\bar{p}} + \frac{\sigma^2}{\bar{p}^3}.$$

Finally, we invert this sum to get $H_K(a)$. Using the approximation $(A + \epsilon)^{-1} \approx A^{-1} - A^{-2}\epsilon$:

$$H_K(a) = \left( \frac{1}{\bar{p}} + \frac{\sigma^2}{\bar{p}^3} \right)^{-1} \approx \bar{p} - \bar{p}^2 \left( \frac{\sigma^2}{\bar{p}^3} \right) = \bar{p}(a) - \frac{\sigma^2(a)}{\bar{p}(a)}. \tag{13}$$

Comparing Eq. equation 12 and Eq. equation 13, we see that maximizing the Infomax objective $J_2$ is equivalent (at the second order) to maximizing the Generalized Harmonic Mean. The variance term $\frac{\sigma^2(a)}{\bar{p}(a)}$ acts as a penalty, discouraging answers where the model confidence fluctuates heavily across views. $\qquad \square$

### G.4 NON-UNIFORM PRIOR AND WEIGHTED INVARIANT SCORE

Now we generalize Theorem 3.2 to the case of a non-uniform prior over views. Let the prior over the views be parameterized by $\pi \in (0, 1)$, such that:

$$p(X = x) = \pi, \qquad p(X = x') = 1 - \pi.$$

The marginal probability of an answer $a$ becomes the weighted average:

$$\bar{p}_\pi(a) = \pi p_0(a) + (1 - \pi)p_1(a).$$

The definition of the Mutual Information $I(Z_a; X)$ generalizes to the weighted sum of KL divergences:

$$I(Z_a; X) = \pi D_{\mathrm{KL}}\big(p_0(a) \parallel \bar{p}_\pi(a)\big) + (1 - \pi)D_{\mathrm{KL}}\big(p_1(a) \parallel \bar{p}_\pi(a)\big). \tag{14}$$

Recall the second-order approximation of the KL divergence $D_{\mathrm{KL}}(p \parallel q) \approx \frac{(p-q)^2}{2q(1-q)}$. We compute the deviations of each view from the weighted marginal. Let $\Delta(a) = p_0(a) - p_1(a)$. Then:

$$p_0(a) - \bar{p}_\pi(a) = p_0(a) - [\pi p_0(a) + (1 - \pi)p_1(a)]$$
$$= (1 - \pi)(p_0(a) - p_1(a)) = (1 - \pi)\Delta(a).$$

$$p_1(a) - \bar{p}_\pi(a) = p_1(a) - [\pi p_0(a) + (1 - \pi)p_1(a)]$$
$$= -\pi(p_0(a) - p_1(a)) = -\pi\Delta(a).$$

Table 8: Comparison between generalized mean and TTRL baseline in fine-grained dataset, Self-Harmony achieves best performance

| Method | Conf. Correct | Conf. Incorrect | Not-Conf. Correct | Not-Conf. Incorrect |
|---|---|---|---|---|
| Arithmetic | 98.83 | 76.95 | 90.82 | 58.13 |
| Harmonic | **99.61** | **77.73** | **93.28** | **61.25** |
| Quadratic | 98.43 | 76.25 | 89.61 | 60.98 |
| Geometry | 99.22 | 76.17 | 91.23 | 58.89 |
| Full Infomax | 99.22 | 75.78 | 90.10 | 60.00 |
| TTRL | 98.90 | 76.81 | 92.35 | 58.97 |

Substituting these into the Taylor expansion of Eq. equation 14:

$$I(Z_a; X) \approx \frac{1}{2\bar{p}_\pi(a)(1 - \bar{p}_\pi(a))} \left[\pi(1 - \pi)^2 \Delta(a)^2 + (1 - \pi)(-\pi)^2 \Delta(a)^2\right]$$

$$= \frac{\Delta(a)^2}{2\bar{p}_\pi(a)(1 - \bar{p}_\pi(a))} \underbrace{\left[\pi(1 - \pi)^2 + \pi^2(1 - \pi)\right]}_{\pi(1-\pi)(1-\pi+\pi)=\pi(1-\pi)}$$

$$= \frac{\pi(1 - \pi)\Delta(a)^2}{2\bar{p}_\pi(a)(1 - \bar{p}_\pi(a))}.$$

Assuming the Balanced-Confidence condition holds, we approximate the denominator $1 - \bar{p}_\pi(a) \approx 1$. The objective function $J_2(a)$ then becomes:

$$J_2(a) \approx \bar{p}_\pi(a) - \frac{\pi(1 - \pi)(p_0(a) - p_1(a))^2}{\bar{p}_\pi(a)}.$$

This Weighted Invariant Score generalizes the Harmonic Mean Selector. When $\pi = 0.5$, the variance term is $\pi(1 - \pi) = 0.25$, recovering the original form. For $\pi \neq 0.5$, the objective balances the *weighted* confidence against the consistency penalty, where the penalty strength is modulated by the prior variance.

## H  EMPIRICAL RESULT FOR GENERALIZED MEAN COMPARISON

We further conducted a fine-grained analysis using Qwen-4B-Base on the Math dataset shown in Table 8. We stratified the problems into four categories based on the base model's inherent confidence and correctness:

- Confident (majority answer $\geq$ N/2 votes) and correct
- Not confident (majority answer $<$ N/2 votes) but correct
- Confident and incorrect
- Not confident and incorrect

Across all four subsets, the harmonic mean yields the best performance, including the most challenging *confident and incorrect* region where models typically hallucinate with high certainty. These results demonstrate that Self-Harmony improves mean pass@1 at 16 rollouts not only in easy high-confidence settings but also under uncertainty and erroneous certainty, highlighting its robustness and general applicability.

## I  INFERENCE-ONLY COMPARISON

To evaluate the harmonic mean as a pseudo-label selector *without any training*, we compare Majority Voting versus Self-Harmony under different inference budgets. Following the *Confident Correct* definition in Section H, we sweep the number of rollouts from 32 to 256 and report Pass@1 (in %) based on the selected answer among the rollouts shown in Table 9.

**Confident-Correct Dataset**

Table 9: Inference-only performance (Pass@1, %) under different rollout budgets. Self-Harmony consistently improves or maintains performance as compute increases, whereas Majority Voting may plateau or degrade.

| Subset | Model | Method | 32 | 64 | 128 | 256 |
|---|---|---|---|---|---|---|
| Confident-Correct | Qwen-4B-Base | Majority Voting | **99.61** | 98.83 | 98.44 | 98.44 |
| Confident-Correct | Qwen-4B-Base | Self-Harmony | **99.61** | **99.61** | **99.61** | **99.22** |
| Confident-Correct | Qwen-8B-Base | Majority Voting | **98.83** | 99.22 | 99.22 | 99.22 |
| Confident-Correct | Qwen-8B-Base | Self-Harmony | **98.83** | **99.61** | **99.61** | **99.61** |
| Confident-Incorrect | Qwen-4B-Base | Majority Voting | 75.78 | 76.95 | 75.78 | 75.39 |
| Confident-Incorrect | Qwen-4B-Base | Self-Harmony | **78.52** | **78.12** | **77.34** | **77.73** |
| Confident-Incorrect | Qwen-8B-Base | Majority Voting | 70.70 | 70.31 | 69.92 | 69.14 |
| Confident-Incorrect | Qwen-8B-Base | Self-Harmony | **71.87** | **71.48** | **70.70** | **71.48** |
| Not-Confident-Correct | Qwen-4B-Base | Majority Voting | 90.23 | 90.23 | 91.40 | 91.40 |
| Not-Confident-Correct | Qwen-4B-Base | Self-Harmony | **95.31** | **94.92** | **94.53** | **94.53** |
| Not-Confident-Correct | Qwen-8B-Base | Majority Voting | 89.45 | 89.84 | 90.62 | 91.01 |
| Not-Confident-Correct | Qwen-8B-Base | Self-Harmony | **94.14** | **94.92** | **94.92** | **95.31** |
| Not-Confident-Incorrect | Qwen-4B-Base | Majority Voting | 64.45 | 64.84 | 66.01 | 65.23 |
| Not-Confident-Incorrect | Qwen-4B-Base | Self-Harmony | **65.62** | **66.41** | **66.80** | **67.19** |
| Not-Confident-Incorrect | Qwen-8B-Base | Majority Voting | 60.93 | 63.28 | 63.67 | 63.28 |
| Not-Confident-Incorrect | Qwen-8B-Base | Self-Harmony | **61.32** | **64.06** | **65.23** | **65.23** |
| AMC | Qwen-4B-Base | Majority Voting | 54.22 | 53.01 | 54.22 | 54.22 |
| AMC | Qwen-4B-Base | Self-Harmony | **56.63** | **56.63** | **57.83** | **57.83** |
| AMC | Qwen-8B-Base | Majority Voting | 54.22 | 56.63 | 55.42 | 56.63 |
| AMC | Qwen-8B-Base | Self-Harmony | **57.83** | **59.04** | **59.04** | **59.04** |

- **Qwen-4B-Base:** Majority Voting slightly decreases as rollouts increase (99.61% → 98.44%), whereas Self-Harmony remains highly stable at ≥ 99.22% across all rollout budgets.

- **Qwen-8B-Base:** Both methods improve with more rollouts, but Self-Harmony consistently converges to the highest accuracy (99.61%) and maintains it from 64 to 256 rollouts, while Majority Voting plateaus at 99.22%.

**Confident-Incorrect Dataset**

- **Qwen-4B-Base:** Self-Harmony outperforms Majority Voting at every rollout setting and remains stable as rollouts increase; Majority Voting fluctuates and never surpasses Self-Harmony.

- **Qwen-8B-Base:** Self-Harmony consistently outperforms Majority Voting across all rollout budgets, while Majority Voting steadily decreases as the number of rollouts increases.

**Not-Confident-Correct Dataset**

- **Qwen-4B-Base:** Majority Voting improves only slightly with more rollouts (90.23% → 91.40%), whereas Self-Harmony delivers a large and consistent gain (+4–5%) and remains highly stable across rollout budgets (95.31% → 94.53%), demonstrating strong correction ability when initial model confidence is low but the answer is correct.

- **Qwen-8B-Base:** Majority Voting improves with additional rollouts but remains far below Self-Harmony at every setting (89.45–91.01% vs. 94.14–95.31%). Self-Harmony quickly saturates at high accuracy and preserves it across budgets, indicating that the method reliably preserves valid low-confidence answers without being swayed by sample variance.

**Not-Confident-Incorrect Dataset**

- **Qwen-4B-Base:** Self-Harmony consistently improves performance as rollouts increase (65.62% → 67.19%), whereas Majority Voting remains stagnant and even decreases at

high rollout counts (64.45% → 65.23%). This shows that Self-Harmony helps correct low-confidence mistakes rather than reinforcing them.

- **Qwen-8B-Base:** Majority Voting shows modest movement with additional rollouts (60.93–63.67%) but never closes the gap with Self-Harmony. In contrast, Self-Harmony scales positively with budget and converges to a much higher accuracy range (64.06–65.23%), indicating that the harmonic selection mechanism is robust against noise when confidence is low and the answer is wrong.

**AMC**

- **Qwen-4B-Base:** Self-Harmony outperforms Majority Voting at all rollout budgets and continues to improve as the sampling budget increases, whereas Majority Voting remains unchanged or fluctuates.

- **Qwen-8B-Base:** Majority Voting improves only modestly with additional rollouts, while Self-Harmony achieves the highest accuracy and sustains it from 64 to 256 rollouts.

**Overall.**  Self-Harmony scales favorably with larger models and can be used in a *plug-and-play* manner at test time to obtain further accuracy gains without any parameter updates or training.

## J  COMPARATIVE SIGNAL QUALITY ACROSS CONFIDENCE–CORRECTNESS QUADRANTS

To pinpoint the source of Self-Harmony's performance gains, we conduct a fine-grained analysis of the training signals. We categorize the base model's initial outputs into four subsets based on confidence and correctness: (1) Confident Correct, (2) Confident Incorrect, (3) Not-Confident Correct, and (4) Not-Confident Incorrect.

As summarized in Table 10, a consistent pattern emerges across all evaluated metrics—Spearman correlation, F1 score, Average Pass@1, and pseudo-label accuracy:

- **Confident Correct:** While both approaches perform well due to saturated correctness in this subset, Self-Harmony demonstrates greater stability (like **99.61%** vs. 98.90% in Pass@1), effectively preventing performance degradation.

- **Confident Incorrect:** This represents the most challenging subset. Self-Harmony achieves *substantial* gains here (like **+21.3%** in Spearman $\rho$), indicating a superior capability to suppress confidently wrong answers that typically mislead Majority Voting.

- **Not-Confident Correct:** Self-Harmony recovers weak-but-correct signals more effectively, yielding cleaner training supervision for ambiguous or borderline questions.

- **Not-Confident Incorrect:** Even when neither confidence nor correctness is reliable, Self-Harmony maintains more discriminative supervision compared to the baseline.

Collectively, these results demonstrate that Self-Harmony's improvement is not derived merely from "easy" samples. Instead, the framework significantly enhances pseudo-label quality in the **critical areas where Majority Voting is fragile**: specifically, confidently wrong and ambiguous instances. This robustness explains why downstream learning benefits more from Self-Harmony than from standard self-consistency aggregation.

## K  INTERPRETATION OF HARMONIC MEAN

$$\text{Harmonic Mean} = \frac{2p_0(a)p_1(a)}{p_0(a) + p_1(a)} = \frac{1}{2}\left[\underbrace{(p_0(a) + p_1(a))}_{\text{Self Consistency}} - \underbrace{\frac{(p_0(a) - p_1(a))^2}{p_0(a) + p_1(a)}}_{\text{View Invariance}}\right]$$

This decomposition provides a clear interpretation of the harmonic mean pseudo-labeling objective. The first term, highlighted in green, rewards high overall confidence from both views — analogous to maximizing the likelihood under view-agnostic aggregation. The second term, highlighted in

Table 10: Performance breakdown across four confidence–correctness quadrants. Self-Harmony consistently outperforms Majority Voting (MV) in the most difficult subsets (Confident Incorrect and Not-Confident Correct) across all metrics.

| Metric | Method | Confident Correct | Confident Incorrect | Not-Confident Correct | Not-Confident Incorrect |
|---|---|---|---|---|---|
| Spearman Correlation | Self-Harmony | 18.35 | **36.51** | **37.16** | **30.11** |
| ($\rho$, %) | Majority Voting | **35.82** | 15.19 | 35.83 | 29.34 |
| F1 Score | Self-Harmony | **99.35** | **75.76** | **93.07** | **66.43** |
| (%) | Majority Voting | 88.89 | 69.87 | 91.26 | 63.41 |
| Pseudo-label | Self-Harmony | **99.22** | **79.69** | **95.31** | **68.75** |
| Accuracy (%) | Majority Voting | 96.09 | 65.62 | 90.62 | 60.16 |
| Avg. Pass@1 | Self-Harmony | **99.61** | **77.73** | **93.28** | **61.25** |
| (16 rollouts, %) | Majority Voting | 98.90 | 76.81 | 92.35 | 58.97 |

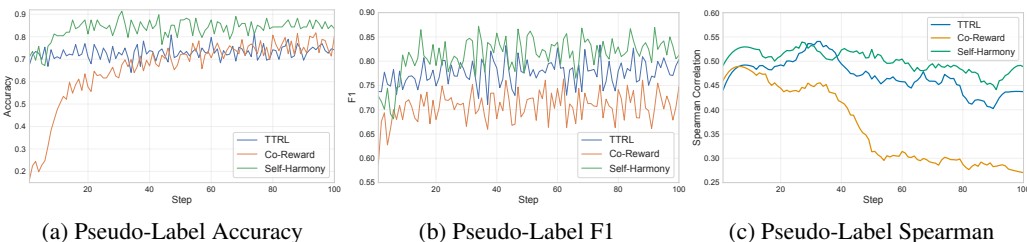

(a) Pseudo-Label Accuracy      (b) Pseudo-Label F1      (c) Pseudo-Label Spearman

Figure 12: Comparison of pseudo-label accuracy, F1, and Smoothed spearman correlation across training steps for TTRL, Co-Reward, and Self-Harmony on Math500 dataset. Self-Harmony provides consistently stronger and more stable training signals than the baselines.

blue, penalizes disagreement between views, acting as a view-invariance regularizer. Thus, the harmonic mean simultaneously prefers labels that are (i) confident according to both views and (ii) consistent across views, rather than disproportionately influenced by a single overconfident but unreliable prediction.

## L   COMPARISON OF TRAINING SIGNALS

Figure 12 compares three key training-signal properties—pseudo-label accuracy, F1, and smoothed Spearman correlation—across optimization steps for TTRL, Co-Reward, and Self-Harmony on Math500 dataset. Across all metrics, Self-Harmony consistently provides the strongest and most stable supervision.

## M   USAGE OF LARGE LANGUAGE MODELS

In preparing this paper, we used Large Language Models only to aid with language polishing, grammar refinement, and improving readability. All research ideas, methodological contributions, experimental design, and analysis were conceived and carried out entirely by the authors. The LLM was not involved in ideation, technical writing of results, or scientific claims. The authors take full responsibility for all content of this paper.

