# OpenReview forum: "SELF-HARMONY: LEARNING TO HARMONIZE SELF-SUPERVISION AND SELF-PLAY IN TEST-TIME REINFORCEMENT LEARNING"
_ICLR.cc/2026/Conference — ICLR 2026 Poster_

### Official Review · Reviewer_RRhh · 2025-10-29

**Soundness:** 2
**Presentation:** 2
**Contribution:** 2
**Rating:** 2
**Confidence:** 4

**Summary:**

This paper presents "self-harmony" an alternative to final-response consistency like self-consistency often used for test time scaling and more importantly for unsupervised supervision at test time. While self-consistency involves temperature sampling multiple solutions and selecting the one corresponding to the majority answer, the authors propose looking at the same problem from "multiple views" done by creating multiple rephrasings of the input query and selecting the answer consistent across these views by using the harmonic mean. The model is then trained on this unsupervised signal and the performance is reported. The authors argue that self-harmony is useful even in cases when self-consistency fails, i.e., model consistently arriving at the wrong answer.

**Strengths:**

- Obtaining unsupervised signals for improving reasoning performance is an important area of work with potential applications in scaling reasoning performance during pretraining or in the wild, where annotations are not available. This paper makes a contribution towards the broader goal.
- The paper is well-structured and easy-to-follow with decent empirical gains in the main resutls as well as some theoretical justifications for the view-invariant info-max objective.

**Weaknesses:**

- Missing connections to previous work in "unsupervised" training signals for reasoning:
    - https://arxiv.org/abs/2411.04109
    - https://arxiv.org/abs/2210.11610
    - other papers on RL with external reward models for reasoning: https://arxiv.org/abs/2406.16838, https://arxiv.org/abs/2403.13787, etc.
- While the main motivation hinges on self-consistency being ineffective when the model is consistently wrong, the citations and experiments to back this up are lacking:
    - To the best of my understanding, Liu et al 2025b uses self-consistency to compare across different prompting strategies and the shortcoming of self-consistency at large was not clear to me, the authors need to be more explicit when making this connection (L157-160).
    - Since the work requires actual model training on this unsupervised signal, simply the accuracy of the pseudo label does not suffice (as in Fig 3b). Unclear if the original test set (ground truth labels) is biased towards getting correctness right, or are we measuring the how often the consistency indicator lines up with correct or incorrect prediction. It would be prudent to report F1, or correlation between the metrics and the binary ground truth by Spearman or Somer's D correlation as done in https://arxiv.org/abs/2411.04109.
    - As shown in previous works, https://arxiv.org/abs/2411.04109 and https://arxiv.org/abs/2509.06870 (concurrent), as the size of the majority answer increases, self-consistency becomes a better correctness indicator. Therefore, the authors should show that self-harmony outperforms self-consistency when setting consistency threshold to filter out smaller answer sizes.
- Also see clarifications on experimental setup underspecified in the paper below.

**Questions:**

In addition to weaknesses, I have the following questions about the experimental setup:
- What is the size of the training dataset, are we to assume that the test set (without labels) was used as the unsupervised signal? Is the same model trained for multiple tasks or different ones, please specify and compare against the other case? If the training is done on test queries (which are rather small -- at least for Math), how does the performance scale as the amount of training data increases?
- Why do the authors use base model for Qwen but instruct models for Lllama?
- Why not have GRPO with a trained reward model (since the test set is mainstream evaluation for reasoning tasks) aimed at reasoning as a baseline too and show self-harmony outperforms it?
- Regarding 3(a), how often does the collapse happen and is it just on this specific dataset and/or model type? What are possible explanations or is it the entropy collapse seen in RLHF and how does the step where it happens relate to the size of the training set and the batch size?
- Previous works show that self-consistency is useful even as the model size scales https://arxiv.org/abs/2509.06870, does the same hold for self-harmony. Can self-harmony be used without training to get more gains on the testsets?
- The main paper should also contain examples of the rephrasings and any evaluation done to ensure the content and original intent were preserved.

---

> ### Author Response · Authors · 2025-11-21
> **Response to Reviewer RRhh (Part 1/5)**
>
> We thank the reviewer‘s constructive feedback. We are encouraged that the reviewer finds the problem of unsupervised signal generation important, the paper well-structured, and the empirical gains and theoretical justifications decent.
>
> We will address reviewer's questions one by one.
>
> ## 1. **Missing connections to previous work in "unsupervised" training signals for reasoning**
> We appreciate the suggested citations and already incorporated in our revision.
> - Regarding [2411.04109] and [2210.11610] (unsupervised signals): These are relevant, and we’ve added them to our Related Work section in  revision to broaden the context on unsupervised signal generation.
> - Regarding [2406.16838] and [2403.13787] (RL with external reward models): We respectfully argue these are a different paradigm. A core contribution of Self-Harmony is being a **fully self-contained, label-free TTRL framework that requires no external models or supervision**. The cited papers rely on *external* reward models, which we explicitly aim to avoid.
> But we have cited these work in revision.
>
> ## 2. **To the best of my understanding, Liu et al 2025b uses self-consistency to compare across different prompting strategies and the shortcoming of self-consistency at large was not clear to me, the authors need to be more explicit when making this connection (L157-160).**
> The reviewer is correct that the main focus of Liu et al. (2025b) is to compare prompting strategies.
> However, we cited this paper specifically for its valuable formal definition of the exact failure mode of self-consistency that motivates Self-Harmony.
> Our statement "...if p(Correct | x) < p(Wrong | x)..." is a direct, plain-English summary of what Liu et al. (2025b) formally define as a "hard question" (Definition 4.1). This is a case where the correct answer $a_{1}$ is not in the set of most probable answers ($\mathcal{S}$), meaning its probability is lower than that of the most-likely incorrect answer.
> Crucially, Theorem 3 from Liu et al. (2025b) then proves the consequence: for these "hard questions," the probability of majority voting (SC) finding the correct answer approaches 0 as the number of samples increases.
> This is the mathematical proof that majority voting "will amplify these spurious solutions."
> Our paper is built to solve this exact, formally-defined problem. In the revision, we've made more discussion on this connection.

---

> ### Author Response · Authors · 2025-11-21
> **Response to Reviewer RRhh (Part 2/5)**
>
> ## 3. **Since the work requires actual model training on this unsupervised signal, simply the accuracy of the pseudo label does not suffice (as in Fig 3b). Unclear if the original test set (ground truth labels) is biased towards getting correctness right, or are we measuring the how often the consistency indicator lines up with correct or incorrect prediction. It would be prudent to report F1, or correlation between the metrics and the binary ground truth by Spearman or Somer's D correlation as done in https://arxiv.org/abs/2411.04109.**
> We thank the reviewer for this valuable suggestion. To address this, we expanded our evaluation on Qwen-4B-Base to include **F1 Score** and **Spearman’s Rank Correlation following your suggestions,**
>
> We analyzed performance across four subsets of problem difficulty: (1) **Confident Correct (where the original model’s** majority answer ≥ N/2 votes is correct **)**, (2) **Confident Incorrect (where the original model’s** majority answer ≥ N/2 votes is **incorrect )**, (3) **Not Confident Correct (where the original model’s** majority answer < N/2 votes is **correct )**, and (4) **Not Confident Incorrect**.
> | Method/Qwen-4B-Base/Average Pass@1 over 16 rollouts | confident correct datasets | confident incorrect datasets | not confident correct datasets | not confident incorrect datasets |
> | --- | --- | --- | --- | --- |
> | harmonic(Self-Harmony) | **99.61** | **77.73** | **93.28** | **61.25** |
> | TTRL(Majority Voting) | 98.90 | 76.81 | 92.35 | 58.97 |
>
> | Method/Qwen-4B-Base/F1 | confident correct datasets | confident incorrect datasets | not confident correct datasets | not confident incorrect datasets |
> | --- | --- | --- | --- | --- |
> | harmonic(Self-Harmony) | **99.35** | **75.76** | **93.07** | **66.43** |
> | TTRL(Majority Voting) | 88.89 | 69.87 | 91.26 | 63.41 |
>
> Self-Harmony achieves significantly higher F1 scores across all subsets. Most notably, in the "Confident Correct" setting, Self-Harmony improves F1 by over 10 points (99.35 vs. 88.89). This indicates that our Harmonic Mean Score (HMS) is far more effective at correctly identifying and retrieving valid training signals without discarding them (high recall) or accepting noise (high precision).
>
> | Method/Qwen-4B-Base/Spearman $\rho$(%) | confident correct datasets | confident incorrect datasets | not confident correct datasets | not confident incorrect datasets |
> | --- | --- | --- | --- | --- |
> | harmonic(Self-Harmony) | 18.35 | **36.51** | **37.16** | **30.11** |
> | TTRL(Majority Voting) | **35.82** | 15.19 | 35.83 | 29.34 |
>
> The most critical finding lies in the Confident Incorrect subset. Here, Majority Voting exhibits a very low correlation (15.19), confirming that when the model hallucinates confidently, vote counts become decoupled from correctness. In contrast, Self-Harmony maintains a much stronger correlation (36.51).
>
> Regarding the Confident–Correct subset, the relatively lower Spearman correlation of Self-Harmony should not be interpreted as reduced pseudo-label quality. Rather, Self-Harmony produces near-perfect consistency in this regime (F1 ≈ 1), leaving very little variability for rank-based correlation metrics to detect. Thus, the drop reflects a statistical ceiling effect rather than a failure of the method.
>
> These results confirm that **HMS successfully penalizes "confident hallucinations" (view-dependent errors)**. Even when the model is confidently wrong, the lower harmonic mean score (due to inconsistency across views) better reflects the low probability of correctness, providing a safer and more aligned training signal than majority voting.

---

> ### Author Response · Authors · 2025-11-21
> **Response to Reviewer RRhh (Part 3/5)**
>
> ## 3. **Comparison to stronger SC baseline (filtering by majority size).**
>
> We agree that Self-Consistency (Majority Voting) becomes a stronger indicator of correctness as the consensus size increases. To verify if Self-Harmony retains its advantage in this "high-confidence regime," we conducted an experiment **filtering for samples where the majority answer appears in at least half of the rollouts ($\ge N/2$, we use $\ge 8$ in 16)**.
> We evaluated Qwen-4B-Base on two high-confidence subsets: (1) Confident Correct (High consensus, answer is correct) and (2) Confident Incorrect (High consensus, answer is wrong).
>
> | Method/Qwen-4B-Base/Average Pass@1 over 16 rollouts | confident correct datasets | confident incorrect datasets |
> | --- | --- | --- |
> | harmonic(Self-Harmony) | **99.61** | **77.73** |
> | TTRL(Majority Voting) | 98.90 | 76.81 |
>
> In High-Confidence Correctness dataset, even when the baseline Majority Voting is extremely effective (98.90%), Self-Harmony still extracts marginal gains, pushing performance nearly to perfection (99.61%).
>
> The Confident Incorrect result is particularly significant. These are cases where the model is "confidently wrong" (high majority consistency for an incorrect answer). Majority voting inevitably reinforces these errors. However, Self-Harmony achieves a higher Pass@1 (77.73% vs 76.81%), suggesting that the Harmonic Mean's dual-view constraint can better detect and recover from these high-confidence hallucinations than simple voting filters.
>
>
> ## 5. **What is the size of the training dataset, are we to assume that the test set (without labels) was used as the unsupervised signal? Is the same model trained for multiple tasks or different ones, please specify and compare against the other case? If the training is done on test queries (which are rather small -- at least for Math), how does the performance scale as the amount of training data increases?**
> As Self-Harmony is a Test-Time Reinforcement Learning (TTRL) framework , adaptation occurs "on the fly" using only the unlabeled test problems themselves. The "training data" is the unlabeled test set itself. Our main results (Table 1) train a separate model for each task/dataset. We agree that studying performance scaling with increasing amounts of unlabeled data is an interesting avenue for future work.
>
> We've run an experiment using Qwen3-4B-Base training on Math500(nearly 500 data), Math training set(nearly 7500 data) and DAPO-Math-17k(nearly 1.7M data) with learning rate = 6e-6, batchsize = 256, and evaluating these three models on different evaluation sets at same step. In step 130: Model trained with Math500 shows better performance in Math500, GSM8K, AIME, MMLU-Pro,  and model trained with Math Training Set shows better performance in AMC, GPQA.
> | Training Set / Evaluation Set | Math500 | GSM8K | AIME | AMC | GPQA | MMLU-Pro |
> | --- | --- | --- | --- | --- | --- | --- |
> | Math500 | **79.68** | **92.46** | **18.33** | 45.40 | 36.29 | **70.46** |
> | Math Training Set | 77.15 | 91.80 | 9.16 | **46.91** | **38.03** | 69.00 |
> | DAPO-math-17k | 75.08 | 91.41 | 3.75 | 39.91 | 34.74 | 66.39 |
>
>
>
> ## 6. **Use of base Qwen vs. instruct Llama models.**
>
> The model choices were driven by evaluation stability rather than performance. Qwen3-Base provides a clean training signal for consistency-based methods, while Llama-3.1-3B-Base does not (my methods and baseline all fail to learn). Conversely, Qwen3-Instruct introduces CoT-heavy traces that break comparability due to batch-size collapse. The selected configurations are simply the ones in which all methods can be meaningfully evaluated under the same training regime.
>
> ## 7. **Why not GRPO with a trained reward model as a baseline?**
> This baseline falls outside our problem setting. A "trained reward model" inherently requires a separate, labeled dataset for supervision. The entire premise of our work is label-free TTRL without any external supervision or auxiliary models. We do, however, compare against a "GT-Reward" (Table 1) as an oracle upper-bound, which is the strongest possible reward signal.
>
> ## 8. **Regarding 3(a), how often does the collapse happen and is it just on this specific dataset and/or model type? What are possible explanations or is it the entropy collapse seen in RLHF and how does the step where it happens relate to the size of the training set and the batch size?**
> TTRL exhibits collapse only in the Llama-3.1-8B-Instruct model on the Math and MMLU-Pro datasets. There is entropy collapse, similar to the failure modes in RLHF. The model reward hacks the majority-vote pseudo-label: once biased toward a wrong answer, the majority signal reinforces that error, driving policy entropy to zero and locking the model into a deterministic, incorrect state. As for training step and batchsize, It happens in early stage of training (~20 steps) for both dataset while batchsize for Math is 128 and for MMLU-Pro is 32 respectively.

---

> ### Author Response · Authors · 2025-11-21
> **Response to Reviewer RRhh (Part 4/5)**
>
> ## 9.**Previous works show that self-consistency is useful even as the model size scales https://arxiv.org/abs/2509.06870, does the same hold for self-harmony. Can self-harmony be used without training to get more gains on the testsets?**
>
> To address the question that if self-harmony is useful even as the model size scales, we summarize the results(metric is Average Pass@1 over 16 rollouts as main table) from Table 1 across three model scales below:
> | Model/Dataset | Math500 | GSM8K | AIME 2024 | AMC | GPQA | MMLU-Pro |
> | --- | --- | --- | --- | --- | --- | --- |
> |Qwen3-1.7B-Base | 69.60 | 87.47 | 10.00 | 40.51 | 27.92 | 53.66|
> |Qwen3-4B-Base | 78.50 | 94.31 | 20.00 | 49.40 | 37.06 | 67.68 |
> |Qwen3-8B-Base | **80.00** | **95.45** | **23.33** | **59.04** | **38.07** | **77.68** |
>
> Self-Harmony scales favorably with model capacity, consistently improving performance from 1.7B $\rightarrow$ 4B $\rightarrow$ 8B without diminishing returns.
>
> To address your query about using HMS purely as an inference-time selection mechanism (without gradient updates), we conducted an ablation study on the Qwen3-4B-Base model and Qwen3-8B-Base. We evaluated the stability of the selection mechanism as we scaled the number of rollouts from 32 to 256.
>
> We report Pass@1 for Self-Consistency (Majority Voting) and Self-Harmony under this inference-only setting on three subsets: Confident-Correct, Confident-Incorrect, and AMC.
>
> For the Confident-Correct and Confident-Incorrect subsets, we follow the filtering rule: math questions for which **the base model’s 16-rollout majority vote count is ≥ 8**, and whose majority prediction is correct or incorrect, respectively. The subsets are constructed independently for Qwen3-4B-Base and Qwen3-8B-Base.
>
> ### Inference Only Result(Pass@1  \%)
>
> | Subset              | Model    | Method          | 32 Rollouts | 64 Rollouts | 128 Rollouts | 256 Rollouts |
> |---------------------|----------|-----------------|-------------|-------------|--------------|--------------|
> | Confident-Correct   | Qwen-4B-Base  | Majority Voting | **99.61** | 98.83 | 98.44 | 98.44 |
> | Confident-Correct   | Qwen-4B-Base  | Self-Harmony| **99.61** | **99.61** | **99.61** | **99.22** |
> | Confident-Correct   | Qwen-8B-Base  | Majority Voting | **98.83** | 99.22 | 99.22 | 99.22 |
> | Confident-Correct   | Qwen-8B-Base  | Self-Harmony| **98.83** | **99.61** | **99.61** | **99.61** |
> | Confident-Incorrect | Qwen-4B-Base  | Majority Voting | 75.78 | 76.95 | 75.78 | 75.39 |
> | Confident-Incorrect | Qwen-4B-Base  | Self-Harmony| **78.52** | **78.12** | **77.34** | **77.73** |
> | Confident-Incorrect | Qwen-8B-Base  | Majority Voting | 70.70 | 70.31 | 69.92 | 69.14 |
> | Confident-Incorrect | Qwen-8B-Base  | Self-Harmony| **71.87** | **71.48** | **70.70** | **71.48** |
> | AMC                 | Qwen-4B-Base  | Majority Voting | 54.22 | 53.01 | 54.22 | 54.22 |
> | AMC                 | Qwen-4B-Base  | Self-Harmony| **56.63** | **56.63** | **57.83** | **57.83** |
> | AMC                 | Qwen-8B-Base  | Majority Voting | 54.22 | 56.63 | 55.42 | 56.63 |
> | AMC                 | Qwen-8B-Base  | Self-Harmony| **57.83** | **59.04** | **59.04** | **59.04** |
>
> - Confident-Correct
> 	- Qwen-4B-Base: Majority Voting slightly decreases as rollouts increase (99.61\% → 98.44\%), while Self-Harmony remains more stable at ≥ 99.22\% across all rollout budgets.
> 	- Qwen-8B-Base: Both methods improve with more rollouts, but Self-Harmony consistently converges to the highest accuracy which is 99.61\% and maintains it from 64 to 256 rollouts, whereas Majority Voting plateaus at 99.22\%.
>
> - Confident-Incorrect
> 	 - Qwen-4B-Base: Self-Harmony is higher than Majority Voting at every rollout setting, and remains stable as rollouts increase; Majority Voting fluctuates and never surpasses Self-Harmony.
> 	 - Qwen-8B-Base: Self-Harmony consistently outperforms Majority Voting across all rollout budgets, while Majority Voting steadily decreases as the number of rollouts increases, whereas Self-Harmony remains higher performance.
>
> - AMC
> 	- Qwen-4B-Base: Self-Harmony outperforms Majority Voting at all rollout budgets, and further improves when increasing the sampling budget, whereas Majority Voting remains unchanged or fluctuates.
> 	- Qwen-8B-Base: Majority Voting increases only modestly with additional rollouts, while Self-Harmony reaches the highest accuracy and sustains it from 64 to 256 rollouts.
>
> Overall, Self-Harmony scales favorably with larger models and can be used plug-and-play at test time to obtain further accuracy gains without any parameter updates or training.
>
>
> ## 10.  **Examples of rephrasings in the main paper.**
> We update the examples in main paper Sec4.4, thank reviewer for suggestions.

---

> ### Author Response · Authors · 2025-11-27
> **Response to Reviewer RRhh (Part 5/5)**
>
> We sincerely thank the reviewer for their constructive feedback, which has prompted valuable additional analysis that strengthens our submission. Based on the reviewer’s comments, we conducted additional experiments and provided more rigorous evaluations to validate our method. The results confirm that:
> - The Harmonic Mean Score is superior to simpler majority voting methods in stronger SC baseline (99.61\% vs 98.90\% in confident correct datasets)
> - Our selection mechanism aligns significantly better with ground truth than standard majority voting across rigorous metrics like F1(75.76\% vs 	69.87\% in confident incorrect datasets) and Spearman correlation(36.51\% vs 15.19\% in confident incorrect datasets).
> - Self-Harmony scales favorably with model capacity, consistently improving performance from 1.7B(69.60\%) $\rightarrow$ 4B(78.50\%) $\rightarrow$ 8B(80.00\%) in Math without diminishing returns.
> - Even without training, HMS still exhibits better performance than marjority voting as an inference-time selection mechanism(78.52\% vs 75.78\% in 32 rollouts in confident incorrect datasets)
>
> We hope these additional analyses help address the reviewer’s concerns, and we would be very grateful if the reviewer could kindly let us know whether there remain questions we can clarify further. We are happy to run more experiments or provide more analysis if useful.

---

### Official Review · Reviewer_962B · 2025-10-31

**Soundness:** 3
**Presentation:** 3
**Contribution:** 3
**Rating:** 6
**Confidence:** 3

**Summary:**

This paper introduces Self-Harmony, a framework where a single model plays two roles: a Solver that answers questions and a Reframer that rephrases them. The method aggregates answer frequencies across the original and reframed questions using the harmonic mean, which is shown to outperform baselines such as majority voting on the original questions.

**Strengths:**

**Soundness**: The proposed method is intuitive and straightforward to implement. It builds on a sound intuition — that a correct answer should remain consistent across semantically equivalent but stylistically distinct formulations of the same question. The paper also provides a comparative analysis of pseudo-label quality between Self-Harmony, TTRL, and Co-Reward, demonstrating that Self-Harmony produces the highest-quality pseudo labels.

**Substance**: The experiments are comprehensive, covering multiple model families including Qwen3 and Llama 3.1/3.2. The proposed approach shows consistent performance improvements across these models. Moreover, the ablation studies indicate that the training process is robust to different hyperparameter settings.

**Weaknesses:**

**Missing analysis**: It would strengthen the paper to include a deeper analysis of why Self-Harmony improves pseudo-label quality. For instance, one could perform a simple inference-based categorization: for each question, sample multiple responses and classify questions into four types based on model confidence and correctness:

- Confident (majority answer ≥ N/2 votes) and correct

- Not confident (majority answer < N/2 votes) but correct

- Confident and incorrect

- Not confident and incorrect

Then, apply Self-Harmony method for generating pseudo-label to analyze where the improvements primarily arise — e.g., whether it mainly helps on low-confidence or high-confidence cases, or corrects certain systematic errors.

**Minor issue**: In Table 2, the caption should appear above the table.

**Questions:**

As mentioned in Weakness 1, I’m curious to see a breakdown showing where the improvements primarily originate compared to methods like majority voting.

---

> ### Author Response · Authors · 2025-11-21
> **Response to Reviewer 962B (Part 1/1)**
>
> We thank Reviewer for the positive assessment of our work, specifically regarding the soundness of the intuition and the comprehensive nature of our experiments. We particularly appreciate the insightful suggestion to analyze performance based on confidence and correctness categories. We have performed this analysis, and the results further validate the robustness of Self-Harmony.
>
>
> ## 1. **As mentioned in Weakness 1, I’m curious to see a breakdown showing where the improvements primarily originate compared to methods like majority voting.**
> We followed your specific recommendation to deepen the analysis of why Self-Harmony improves pseudo-label quality.
>
> Experimental Setup:
> Using the qwen-4b-base model, we categorized the math questions of training set into four types based on the initial model outputs (Majority Voting, N = 16, each category contains 256 samples):
> 1. **Confident Correct:** (Majority answer $\ge$ N/2 votes & correct)
> 2. **Confident Incorrect:** (Majority answer $\ge$ N/2 votes & incorrect)
> 3. **Not Confident Correct:** (Majority answer < N/2 votes & correct)
> 4. **Not Confident Incorrect:** (Majority answer < N/2 votes & incorrect)
>
> We then compared pseudo-labels, F1 score, final mean Pass@1 performance using Self-Harmony (Harmonic Mean Score) versus TTRL (Majority Voting) under the same training settings and inference-only performance under different rollouts.
>
> ### Pseudo-label Accuracy
> | Qwen3-4B-Base / Pseudo Label Accuracy | confident correct datasets | confident incorrect datasets | not confident correct datasets | not confident incorrect datasets |
> | --- | --- | --- | --- | --- |
> | harmonic(Self-Harmony) | **99.22** | **79.69** | **95.31** | **68.75** |
> | Majority Voting | 96.09 | 65.62 | 90.62 | 60.16 |
>
> ### F1 Score
> | Method/Qwen-4B-Base/F1 | confident correct datasets | confident incorrect datasets | not confident correct datasets | not confident incorrect datasets |
> | --- | --- | --- | --- | --- |
> | harmonic(Self-Harmony) | **99.35** | **75.76** | **93.07** | **66.43** |
> | Majority Voting | 88.89 | 69.87 | 91.26 | 63.41 |
>
>
> ### Final Mean Pass@1 (16 rollouts) of the trained model
> | Qwen-4B-Base / Mean Pass@1 in 16 Rollouts  | confident correct datasets | confident incorrect datasets | not confident correct datasets | not confident incorrect datasets |
> | --- | --- | --- | --- | --- |
> | harmonic(Self-Harmony) | **99.61** | **77.73** | **93.28** | **61.25** |
> | Majority Voting  | 98.90 | 76.81 | 92.35 | 58.97 |
>
>
> ### Inference-only (without training) selection
> | Subset | Model | Method | 32 Rollouts | 64 Rollouts | 128 Rollouts | 256 Rollouts |
> | --- | --- | --- | --- | --- | --- | --- |
> | Confident-Correct | Qwen-4B-Base | Majority Voting | **99.61** | 98.83 | 98.44 | 98.44 |
> | Confident-Correct | Qwen-4B-Base | Self-Harmony | **99.61** | **99.61** | **99.61** | **99.22** |
> | Confident-Incorrect | Qwen-4B-Base | Majority Voting | 75.78 | 76.95 | 75.78 | 75.39 |
> | Confident-Incorrect | Qwen-4B-Base | Self-Harmony | **78.52** | **78.12** | **77.34** | **77.73** |
> | Not-Confident-correct | Qwen-4B-Base | Majority Voting | 90.23 | 90.23 | 91.40 | 91.40 |
> | Not-Confident-correct | Qwen-4B-Base | Self-Harmony | **95.31** | **94.92** | **94.53** | **94.53** |
> | Not-Confident-Incorrect | Qwen-4B-Base | Majority Voting | 64.45 | 64.84 | 66.01 | 65.23 |
> | Not-Confident-Incorrect | Qwen-4B-Base | Self-Harmony | **65.62** | **66.41** | **66.80** | **67.19** |
>
> Even though majority voting is already strong in the Confident-Correct subset, Self-Harmony still offers additional improvements there; however, the dominant share of the performance advantage arises from the difficult subsets (Confident-Incorrect and Not-Confident), where majority voting tends to fail.
>
> ## 2. **Minor issue**
> Thank you for pointing this out. We have corrected the formatting in the revision so that the caption for Table 2 now appears above the table.
>
> We believe this additional breakdown provides strong evidence that Self-Harmony is not just improving performance on "easy" samples, but provides critical corrections in cases where the model is uncertain or confidently incorrect. We hope this analysis could address your main concern, and we would be happy to provide further clarification if needed.

---

### Official Review · Reviewer_dk94 · 2025-11-01

**Soundness:** 3
**Presentation:** 2
**Contribution:** 3
**Rating:** 4
**Confidence:** 3

**Summary:**

This paper proposes Self-Harmony, a label-free test-time reinforcement learning (TTRL) framework that improves reasoning stability without external supervision. The key idea is that a correct answer should remain consistent across both an original question and its paraphrase. A single model plays two cooperative roles: a Solver that answers and a Reframer that rephrases the question to provide a second view. Pseudo-labels are then chosen using the harmonic mean of answer frequencies from both views, emphasizing solutions robust to rewording while down-weighting view-specific errors. The authors provide theoretical justification for this criterion and integrate it into a self-play reinforcement loop. Across five reasoning benchmarks (MATH500, GSM8K, AIME24, GPQA, MMLU-Pro) and five open-source models, Self-Harmony achieves new state-of-the-art results—ranking first in 28 of 30 settings—and demonstrates superior training stability and pseudo-label accuracy.

**Strengths:**

This paper stands out for its clarity and strong empirical results. The idea of enforcing consistency across paraphrased versions of a question is simple yet powerful—it directly targets a core weakness of existing test-time RL methods that overfit to their own biases. Using a single model in dual roles (Solver and Reframer) is elegant and avoids the need for extra models or supervision, making the approach practical and scalable. The harmonic mean pseudo-label rule is theoretically grounded and nicely bridges information-theoretic reasoning with an intuitive motivation.

The experiments are extensive and convincing: strong gains across five benchmarks, stable training curves, and robustness across model sizes all strengthen the paper’s credibility. Overall, Self-Harmony feels like a solid, well-executed contribution that advances test-time adaptation in a meaningful and reproducible way.

**Weaknesses:**

The harmonic mean rule, while supported by a theoretical derivation, still feels somewhat heuristic in how it’s applied. In practice, the paper does not deeply analyze why the harmonic mean performs better than simpler alternatives like averaging or weighted voting, or under what conditions it might fail. For instance, if the paraphrasing step inadvertently shifts the semantics of the question—introducing small wording biases or contextual cues—the harmonic mean could incorrectly down-weight valid answers. A sensitivity analysis or qualitative study of such “semantic drift” cases would make the claim of robustness more convincing.

Finally, the “self-contained” assumption might break for smaller or less instruction-tuned models, since paraphrasing quality heavily depends on the model’s own linguistic competence—an issue not fully explored in the experiments.

**Questions:**

Could you provide more intuition or empirical evidence for why the harmonic mean is superior to arithmetic or geometric averaging in this setting?

How sensitive is the performance to small perturbations in the paraphrased questions or to imperfect semantic equivalence?

---

> ### Author Response · Authors · 2025-11-21
> **Response to Reviewer dk94 (Part 1/2)**
>
> We thank the reviewer for the constructive feedback and for recognizing **Self-Harmony** as a "simple yet powerful" idea with "strong empirical results" and "superior training stability." We appreciate the acknowledgment that our method achieves SOTA in 28/30 settings.
>
> We provide the additional response for addressing the primary concerns regarding the superiority of the Harmonic Mean over other pseudo-label alternatives and robustness to semantic drift.
>
> ## 1. **Could you provide more intuition or empirical evidence for why the harmonic mean is superior to arithmetic or geometric averaging in this setting?**
>
> Thanks for your suggestion. While Section 4.3.2 validates Self-Harmony against Majority Voting, we acknowledge the need to explicitly compare the our Harmonic Mean Score (HMS) against other aggregation methods (e.g., Arithmetic and Geometric means) to understand their behaviors under different conditions.
>
> To address this, we conducted a fine-grained analysis using Qwen-4B-Base on the math dataset. We stratified the problems into four categories based on the base model’s inherent confidence and correctness:
>
> - Confident (majority answer ≥ N/2 votes) and correct
> - Not confident (majority answer < N/2 votes) but correct
> - Confident and incorrect
> - Not confident and incorrect
>
> The following tables report the Mean Pass@1 (over 16 rollouts), Pseudo-Label Accuracy, and F1 scores across these categories.
>
> Qwen3-4B-Base/Mean Pass@1 | confident correct datasets | confident incorrect datasets | not confident correct datasets | not confident incorrect datasets |
> | --- | --- | --- | --- | --- |
> | arithmetic mean  | 98.83 | 76.95 | 90.82 | 58.13 |
> | harmonic mean (Self-Harmony) | **99.61** | **77.73** | **93.28** | **61.25** |
> | quadratic mean | 98.43 | 76.25 | 89.61 | 60.98 |
> | geometry mean | 99.22 | 76.17 | 91.23 | 58.89 |
> | TTRL(majority voting) | 98.90 | 76.81 | 92.35 | 58.97 |
>
>
> | Qwen3-4B-Base / Pseudo Label Accuracy | confident correct datasets | confident incorrect datasets | not confident correct datasets | not confident incorrect datasets |
> | --- | --- | --- | --- | --- |
> | arithmetic | **99.22** | 77.34 | 94.53 | 67.19 |
> | harmonic(Self-Harmony) | **99.22** | **79.69** | **95.31** | **68.75** |
> | quadratic | **99.22** | 78.91 | 93.75 | 66.40 |
> | geometry | **99.22** | 78.91 | 94.53 | 67.19 |
>
> | Method/Qwen-4B-Base/F1 | confident correct datasets | confident incorrect datasets | not confident correct datasets | not confident incorrect datasets |
> | --- | --- | --- | --- | --- |
> | arithmetic | 99.08 | 71.52 | 90.86 | 57.64 |
> | harmonic(Self-Harmony) | **99.35** | **75.76** | **93.07** | **66.43** |
> | quadratic | 99.08 | 71.85 | 90.60 | 64.09 |
> | geometry | 98.43 | 71.81 | 90.51 | 64.21 |
>
> We could see **Self-Harmony (HMS) consistently outperforms all simpler alternatives across every category** and In the "Confident Incorrect" category, HMS surpassing Arithmetic Mean and Geometric Mean. This demonstrates that even when the base model exhibits strong bias toward a wrong answer, the strict consistency requirement of the Harmonic Mean allows it to better recover the correct solution via the reframed view. This is direct evidence to why the harmonic mean is superior to arithmetic or geometric averaging in this setting.
>
> These results empirically align with our theoretical analysis (Section 3.2) and Appendix G.2 in the revision, where we demonstrate that the tunable parameter $\lambda$ correlates with different values of $k$ in generalized means. Unlike Arithmetic averaging, which allows high confidence in one view to compensate for low confidence in another, HMS penalizes inconsistency. This property makes it the most effective "filter" for selecting high-quality pseudo-labels, particularly in challenging scenarios where the model's raw confidence is unreliable.

---

> ### Author Response · Authors · 2025-11-21
> **Response to Reviewer dk94 (Part 2/2)**
>
> ## 2. **How sensitive is the performance to small perturbations in the paraphrased questions or to imperfect semantic equivalence?  target on “introducing small wording biases or contextual cues—the harmonic mean could incorrectly down-weight valid answers”**
>
> We appreciate this insightful concern. The reviewer identifies a potential risk: if the paraphrasing step introduces semantic drift or wording biases, the strict Harmonic Mean criterion might incorrectly down-weight valid answers.
>
> To quantify this risk, we conducted a safety stress test with the Confident Correct subset of the dataset. On this subset, the Initial Model has a high baseline reliability (Mean Pass@1 $\approx$ 84-89%). If semantic drift were a major issue, the Harmonic Mean would punish these valid answers, causing the mean Pass@1 score in 16 rollouts to drop. **To simulate imperfect semantic equivalence, we utilized model scale as a proxy for paraphrase quality**. We tested Solvers of varying sizes paired with Reframers ranging from a tiny 0.6B (high noise/perturbation) to 14B.
>
> | Self-Harmony/confident correct datasets | Baseline | Ours with 0.6b-reframer | Ours with 1.7b-reframer | Ours with 4b-reframer | Ours with 14b-reframer |
> | --- | --- | --- | --- | --- | --- |
> | Qwen3-1.7B-Base | 86.32 | 94.92 | 94.92 | 95.31 | 96.48 |
> | Qwen3-4B-Base | 83.98 | 96.48 | 97.65 | 98.04 | 98.36 |
> | Qwen3-8B-Base | 89.06 | 97.85 | 97.75 | 98.02 | 99.07 |
>
> - Robustness to Semantic Drift: Even when using a 0.6B Reframer which generates lower-fidelity paraphrases and introduces significant wording bias, we observe massive gains in Mean Pass@1 over the Initial Model (like Qwen-4B improves from 83.98% $\to$ 96.48%). This confirms that the Harmonic Mean is not overly sensitive to perturbations; it successfully identifies consistency even when the rephrasing is noisy, without discarding valid answers.
> - Effectiveness on Small Models: Addressing the concern regarding the "self-contained" assumption for smaller models, our method functions exceptionally well even with a 1.7B Solver (86.32% $\to$ 94.92%). This contradicts the worry that high linguistic competence is a strict prerequisite; the method is robust across model scales.
>
> We hope these additional experiments could provide sufficient clarity to reviewer's concern, and we would be grateful to know if there are any remaining questions we can further clarify. We look forward to continuing the discussion.

---

### Official Review · Reviewer_v4TH · 2025-11-02

**Soundness:** 3
**Presentation:** 3
**Contribution:** 3
**Rating:** 8
**Confidence:** 3

**Summary:**

The authors propose **Self-Harmony**, a new TTRL framework inspired by the Infomax principle. The core intuition is that the correct answer should appear consistently (robustly) across different versions of the question. The authors demonstrate that the argmax of the harmonic mean of the two probabilities under two different versions of the question approximately yields a good pseudo-label, and this is algorithmically achieved via appropriate prompt-engineered self-play. Its empirical efficacy is shown across various reasoning benchmarks and Qwen & Llama models, showing its superiority over prior TTRL approaches.

**Strengths:**

- Well-written
- Clear and solid intuition
- Theoretically well-grounded
- Extensive experiments and benchmarks, as well as publicly released anonymized codes. Ablations provide a complete understanding of each component of the proposed algorithm and clearly show its superiority.

**Weaknesses:**

- The authors mention that the reward design plays a role in obtaninig SOTA performance (Section 4.3), but the only mention of how the weights involved with the rewards are set ($w_f, w_d$) is in Appendix E.4, mentioned in the passing only "weights set to match the proportional influence ...". This design choice should sufficiently be elaborated as well.
- Although there isn't anything (significantly) wrong with the theoretical discussions, I believe there are some parts that warrant further discussion:
  - I think it is beneficial to explicitly state the "second-order approximation" in the main text for the camera-ready version.
  - The assumption that the view random variable $V$ is uniform is implicit and made explicitly only in the proof. This should be made clear in the main text.
  - The theory is only derived for two views and $\lambda = 2$ and uniform prior. I believe that more discussions on this would make the paper better; see Questions.

**Questions:**

1. Is there any connection between the core intuition here and group-invariant learning [1] or invariant risk minimization [2]? I don't recall any discussions relating to this line of works in the paper.

2. (Minor) Are there any literature from human psychology or education for the claim "common human robustness heuristic: when confronted with uncertainty, people often check solutions ..." (line 161)?

3. What happens if $\lambda$ is also tunable? In other words, under two views, would one still be able to derive Eqn. (1) as a function of $\lambda$? If that is the case, then what is the impact of tuning $\lambda$ across various values?

4. Similarly, with fixed $\lambda$, what happens if one considers more than two views? For instance, when $\lambda = 2$, then does multiple views lead to the general form of the harmonic mean? What might be the impact of having more views? At least from theoretical viewpoint, more (diverse) views should give better and better performance, although there is a possibility of plateauing over certain threshold.

5. Follow-up question: If it is possible, I wonder what the impact of the second-order approximation is. Yes, computationally, having a closed-form solution is hugely beneficial, but the second-order approximation of the KL is tight only when the two probabilities are quite similar. Thus, if one puts in enough budget, I wonder if maximizing the infomax objective directly (with an appropriate KL estimator) gives better pseudo-label, and thus better performance. Or, whether the performance gain is minimal and the approximation objective is good as it is.

6. Also a follow-up: the authors consider uniform prior over the views. I wonder if one could be a bit more clever here, e.g., depending on the "persona" or the "strengths" of the current LLM, one could take that into account to make the prior non-uniform?

7. (Minor) As in many infomax papers (and as a theoretician myself), I wonder if this formulation of infomax (and its approximation) has been theoretically studied before, i.e., whether prior generalization bounds are applicable here (e.g., [3]). Or does this warrant further investigation for future work? If yes, which part of the new formulation yields novelty compared to prior literature?

8. (Minor) Very recently, there have been some issues regarding floating-point operations (https://x.com/QPHutu/status/1984258808332550245). Can the authors comment on whether this issue(?) applies to their experiments?



[1] https://www.jmlr.org/papers/v21/20-163.html

[2] https://arxiv.org/abs/1907.02893

[3] https://www.sciencedirect.com/science/article/abs/pii/S1566253525008383 (arXiv: https://arxiv.org/abs/2501.16768)

---

> ### Author Response · Authors · 2025-11-21
> **Response to Reviewer v4TH (Part1/5)**
>
> We sincerely thank Reviewer for exceptionally positive and constructive review, and for the high rating of 8 (accept). We are delighted that they found our paper "well-written," with "clear and solid intuition," "theoretically well-grounded," and supported by "extensive experiments."
>
> Their thoughtful questions and suggestions are invaluable, and we will incorporate them to strengthen the revision. We address each point below.
>
> ## 1. **The authors mention that the reward design plays a role in obtaninig SOTA performance (Section 4.3), but the only mention of how the weights involved with the rewards are set $(w_f, w_d)$ is in Appendix E.4, mentioned in the passing only "weights set to match the proportional influence ...". This design choice should sufficiently be elaborated as well.**
>
> We wish to clarify that the weights for the additive baseline were not chosen arbitrarily; they were mathematically derived to ensure a rigorous and fair comparison with our multiplicative design. Specifically, we aligned the weights based on the effective gradients derived from a first-order Taylor expansion.
>
> Our multiplicative reward is defined as:
> $$R_{\text{mult}} = (1 - w_f R_f)(1 - w_d R_d)R_{\text{solve}}$$
> For correct answers ($R_{\text{solve}}=1$) and small penalty terms, this linearizes to:
> $$R_{\text{mult}} \approx R_{\text{solve}} - w_f R_f - w_d R_d$$
> To control for hyperparameter influence, we configured the additive baseline ($R_{\text{add}} = w_3 R_{\text{solve}} + w_1 R_f + w_2 R_d$) to match this approximation strictly:
> $$w_3 = 1, w_1 = -w_f, w_2 = -w_d$$
>
> This alignment ensures that both models are subject to identical penalty magnitudes for format and diversity violations. Consequently, the superior performance of Self-Harmony can be attributed to the structural advantage of the multiplicative design (which acts as a strict "AND" gate) rather than disparate hyperparameter settings.
>
> ## 2. **Although there isn't anything (significantly) wrong with the theoretical discussions, I believe there are some parts that warrant further discussion:**
> -  **I think it is beneficial to explicitly state the "second-order approximation" in the main text for the camera-ready version.**
>
>     Thank you for pointing out. We’ve updated second-order approximation in revision already.
>
> - **The assumption that the view random variable is uniform is implicit and made explicitly only in the proof. This should be made clear in the main text.**
>
>     Thank you for pointing out. We’ve updated approximation in revision already.
>
> ## 3. **Is there any connection between the core intuition here and group-invariant learning[1] or invariant risk minimization[2]? I don't recall any discussions relating to this line of works in the paper.**
>
> We thank the reviewer for this insightful comment. Our approach is indeed conceptually related to group-invariant learning [1] and invariant risk minimization (IRM) [2], and we made this connection explicit in the revision.
>
> Relation to group-invariant learning.
>
> Paraphrased questions can be interpreted as task-specific transformations that preserve semantic meaning. Assumption 3.1 states that correctness of the true answer is approximately invariant under these transformations, whereas spurious answers are view-dependent. While we do not build a group-equivariant architecture in the strict sense of [1], the View-Invariant Infomax objective operationalizes a similar invariance principle—applied at the level of answers under paraphrased views rather than features under group actions.
>
> Relation to IRM.
>
> IRM enforces predictors that perform optimally across multiple environments. In our setting, paraphrased views $X \in \{x, x'\}$ serve as environments, and Assumption 3.1 posits that only the true answer remains environment-invariant. The objective
> $J_\lambda(a) = I(Z_a; A) - \lambda\, I(Z_a; X)$ maximizes correctness while penalizing dependence on view-specific information, mirroring the IRM constraint but at test-time pseudo-label selection rather than representation learning during supervised training.
>
> We appreciate the reviewer for highlighting this theoretical angle; it helped us better position our contribution within the broader invariance literature.

---

> ### Author Response · Authors · 2025-11-21
> **Response to Reviewer v4TH (Part 2/5)**
>
> ## 4. **(Minor) Are there any literature from human psychology or education for the claim "common human robustness heuristic: when confronted with uncertainty, people often check solutions ..." (line 161)?**
>
> The reviewer is correct that the intuition has precedents in human psychology and education.
>
> (i) Polya’s problem-solving heuristics. In How to Solve It (Polya, 1945), one of the core verification strategies is: “Can you obtain the result differently?” — i.e., when a solver is uncertain, they intentionally reformulate the problem to validate the answer from another perspective. This directly aligns with our mechanism of reframing under uncertainty.
>
> (ii) Cognitive Flexibility Theory. Spiro et al. (1988, 1992) show that in ill-structured problems, reasoning through multiple representations and revisiting content from alternative perspectives enables more robust judgments than relying on a single formulation. While originally framed for learning rather than answer verification, it supports the broader psychological intuition that multiple perspectives help mitigate uncertainty.
>
> We cite these work in revision already.
>
> ## 5. **What happens if is also tunable? In other words, under two views, would one still be able to derive Eqn. (1) as a function of ? If that is the case, then what is the impact of tuning across various values?**
>
> We thank the reviewer for this insightful question. We confirm that **yes**, Eqn. (1) can be derived as a function of an arbitrary λ, and doing so reveals a fascinating connection between the Invariant Infomax objective and the family of Generalized Means.
>
> ### 1. Deriving the General Form
>
> Following the proof in Appendix A (specifically Eq. 13 and 14), we derived the second-order Taylor approximation for the view-consistency term:
>
> $$
> I(Z_a; X) \approx \frac{\delta(a)^2}{2\bar{p}(a)}
> $$
>
> If we leave λ as a free parameter in the objective $J_λ(a) = I(Z_a; A) - λ I(Z_a; X)$, the objective approximates to:
>
> $$
> J_\lambda(a) \approx \bar{p}(a) - \lambda \frac{\delta(a)^2}{2\bar{p}(a)}
> $$
>
> ### 2. Connection to Generalized Means
>
> To understand the impact of tuning λ, we compare $J_λ(a)$ to the Taylor expansion of the Generalized Mean with exponent *k*, denoted as:
>
> $$
> M_k(p_0, p_1) = \left(\frac{p_0^k + p_1^k}{2}\right)^{1/k}
> $$
>
> The second-order expansion of $M_k$ around the mean $\bar{p}$ is known to be:
>
> $$
> M_k(p_0, p_1) \approx \bar{p}(a) + (k-1)\frac{\delta(a)^2}{2\bar{p}(a)}
> $$
>
> By equating the coefficients of the variance term in $J_λ$ and $M_k$, we find a direct mapping between the penalty weight λ and the power mean exponent *k*:
>
> $$
> -\frac{\lambda}{2} = \frac{k-1}{2} \quad \implies \quad k = 1 - \lambda
> $$
>
> ### 3. Impact of Tuning λ
>
> Tuning λ effectively changes which Pythagorean mean the objective maximizes. Increasing λ increases the penalty for disagreement between views (variance), making the selector more "conservative."
>
> - **λ = -1 ⟹ k = 2 (Quadratic Mean/RMS):** The objective maximizes $\sqrt{(p_{0}^2 + p_{1}^2)/2}$. This effectively adds a bonus for variance ($\approx \bar{p} + \frac{\delta^2}{2\bar{p}}$). It is "risk-seeking": it rewards cases where views strongly disagree (high variance), which is generally undesirable for consistency checking.
> - **λ = 0 ⟹ k = 1 (Arithmetic Mean):** The objective maximizes $\bar{p}(a)$. This corresponds to standard Majority Voting or mean-probability selection, which ignores view consistency entirely (coefficient of $\delta^2$ is 0).
> - **λ = 1 ⟹ k = 0 (Geometric Mean):** The objective maximizes $\sqrt{p_{0}(a)p_{1}(a)}$. This balances confidence and consistency moderately.
> - **λ = 2 ⟹ k = -1 (Harmonic Mean - Our Proposal):** As derived in Theorem 1, this maximizes $\frac{2p₀p₁}{p₀ + p₁}$. It penalizes variance with a coefficient of $-1$ ($\approx \bar{p} - \frac{\delta^2}{2\bar{p}}$).
> - **λ → ∞ ⟹ k → -∞ (Minimum):** The objective approaches maximizing $min(p_0, p_1)$, the most conservative selection strategy.
>
> With experiment showing below, the **harmonic mean is the best choice for pseudo label selector for our TTRL settings**. we evaluated the selectors on the Qwen3-4B-Base model. We categorized questions into four types based on model confidence (majority vote > N(number of rollouts)/2) and correctness(majority vote answer = ground truth).
>
> | Qwen3-4B-Base | confident correct dataset | confident incorrect dataset | not confident correct dataset | not confident incorrect dataset|
> | --- | --- | --- | --- | --- |
> | arithmetic | 98.83 | 76.95 | 90.82 | 58.13 |
> | harmonic(Self-Harmony) | **99.61** | **77.73** | **93.28** | **61.25** |
> | quadratic | 98.43 | 76.25 | 89.61 | 60.98 |
> | geometry | 99.22 | 76.17 | 91.23 | 58.89 |

---

> ### Author Response · Authors · 2025-11-21
> **Response to Reviewer v4TH (Part 3/5)**
>
> ## 6. **Similarly, with fixed λ, what happens if one considers more than two views? For instance, when K > 2, does multiple views lead to the general form of the harmonic mean? What might be the impact of having more views? At least from theoretical viewpoint, more (diverse) views should give better and better performance, although there is a possibility of plateauing over certain threshold.**
> We thank the reviewer for this insightful comment. You have correctly identified a natural extension of our theoretical framework.
>
> ### Generalization to Multiple Views (K > 2)
>
> You asked if considering more than two views leads to the general form of the harmonic mean. The answer is **yes** as well.
>
> We have derived the extension of Theorem 3.1 for K views. Following the same logic as our proof for K=2, we assume X is uniformly distributed over K views {$x_1, …, x_k$}. Let $\bar{p}(a)$ be the arithmetic mean and σ²(a) be the variance across views:
>
> $$
> p̄(a) = \frac{1}{K}\sum_{k=1}^{K} p(a|x_k)
> $$
>
> $$
> σ²(a) = \frac{1}{K}\sum_{k=1}^{K} (p(a|x_k) - \bar{p}(a))²
> $$
>
> Using the second-order Taylor expansion of the KL-divergence, the mutual information term generalizes to the variance:
>
> $$
> I(Z_a; X) = \frac{1}{K}\sum_{k=1}^{K} \mathrm{KL}(p_k \| \bar{p}) \approx \frac{σ²(a)}{2\bar{p}(a)(1-\bar{p}(a))}
> $$
>
> Applying the same simplification as the main proof (approximating the denominator 1-\bar{p} ≈ 1), our objective with λ=2 becomes:
>
> $$
> J_2(a) \approx \bar{p}(a) - \frac{σ^2(a)}{\bar{p}(a)}
> $$
>
> Separately, if we take the **Generalized Harmonic Mean** of K views, and perform a Taylor expansion around the arithmetic mean p̄, we obtain the exact same form:
>
> $$
> H_K(a) = \left(\frac{1}{K} \sum_{k=1}^{K} p(a|x_k)^{-1}\right)^{-1}
> $$
>
> $$
> H_K(a) \approx \bar{p}(a) - \frac{σ²(a)}{\bar{p}(a)}
> $$
>
> Thus, maximizing the invariant Infomax objective with λ=2 for K views is theoretically equivalent to maximizing the Generalized Honic Mean.
>
> ### Impact of More Views and Plateauing
>
> Regarding the impact of increasing K, we agree with your hypothesis:
>
> • **Performance:** Theoretically, increasing K provides a better estimate of the "true" probability by reducing the variance of the estimator, leading to better separation between correct and incorrect pseudolabels.
>
> • **Plateauing:** In practice, performance plateaus because generating K *diverse* (independent) views becomes increasingly difficult. As views become correlated (e.g., similar prompts yielding similar logits), the marginal gain in Mutual Information diminishes. Furthermore, if additional views are of lower quality (high noise), they may degrade the consensus.
>
>
>
>
> ## 7. **Follow-up question: If it is possible, I wonder what the impact of the second-order approximation is. Yes, computationally, having a closed-form solution is hugely beneficial, but the second-order approximation of the KL is tight only when the two probabilities are quite similar. Thus, if one puts in enough budget, I wonder if maximizing the infomax objective directly (with an appropriate KL estimator) gives better pseudo-label, and thus better performance. Or, whether the performance gain is minimal and the approximation objective is good as it is.**
>
> To evaluate whether the second-order approximation limits performance, we implemented a KL-based estimator and optimized the full Infomax objective without approximation. Specifically, we replaced our quadratic surrogate with the divergence term.
> $g(p)=\mathrm{KL}\bigl(\mathrm{Bern}(p)\,\Vert\,\mathrm{Bern}(q)\bigr)
> =p\ln\frac{p}{q}+(1-p)\ln\frac{1-p}{1-q}$
> and compared the resulting pseudo-labels against Self-Harmony. The evaluation metric is mean pass@1 across 16 rollouts, stratified into the four difficulty subsets:
>
> | Qwen3-4B-Base | confident correct dataset | confident incorrect dataset | not confident correct dataset | not confident incorrect dataset|
> | --- | --- | --- | --- | --- |
> | harmonic(Self-Harmony) | **99.61** | **77.73** | **93.28** | **61.25** |
> | full infomax | 99.22 | 75.78 | 90.10 | 60.00 |
>
> Directly optimizing the full Infomax objective does not improve over the second-order approximation; in fact, Self-Harmony achieves the highest performance in all subsets. Since Self-Harmony corresponds to the second-order approximation of the Infomax objective, this indicates that the approximation is not only computationally efficient but also advantageous in practice. The second-order formulation naturally suppresses such high-order fluctuations, preventing the fitting of high-order noise while still rewarding consistency across paraphrased views.

---

> ### Author Response · Authors · 2025-11-21
> **Response to Reviewer v4TH (Part 4/5)**
>
> ## 8. **Also a follow-up: the authors consider uniform prior over the views. I wonder if one could be a bit more clever here, e.g., depending on the "persona" or the "strengths" of the current LLM, one could take that into account to make the prior non-uniform?**
>
> We thank the reviewer for this insightful suggestion. We utilized a uniform prior to model the "maximum entropy" scenario where the agent has no a priori knowledge of which view (or persona) is more reliable. This ensures the selector remains parameter-free and robust.
> However, we agree that if one could estimate "persona strength" (e.g., via confidence scores), a non-uniform prior is the theoretically correct extension. One can indeed be "cleverer" by incorporating a prior $\pi \in (0,1)$ for the first view. This generalizes our theorem as follows:
> - Weighted Marginal: The consensus probability becomes the weighted average $\bar{p}\_{\pi(a)} = \pi p\_{0}(a) + (1-\pi ) p\_{1}(a)$
> - Weighted Penalty: The consistency term $I(Z_a;X)$ becomes a weighted sum of KL-divergences: $I(Z_a;X) = \pi KL(p\_0 \parallel \bar{p}\_{\pi}) + (1-\pi) KL(p\_1 \parallel \bar{p}\_{\pi})$
> - Resulting Objective: Applying the second-order Taylor expansion yields a penalty proportional to the prior's variance $\pi(1-\pi)$. The resulting Weighted Invariant Score is: $S\_{\pi(a)} \approx \bar{p}\_{\pi(a)} - \frac{\pi(1-\pi)(p\_{0}(a) - p\_{1}(a))^2}{\bar{p}\_{\pi(a)}}$
>
> This confirms the reviewer's intuition: if we identify a stronger persona (e.g., $\pi \to 1$), the variance term $\pi(1-\pi)$ shrinks, effectively relaxing the consistency constraint and allowing the stronger view to dominate.
>
> ## 9. **(Minor) As in many infomax papers (and as a theoretician myself), I wonder if this formulation of infomax (and its approximation) has been theoretically studied before, i.e., whether prior generalization bounds are applicable here (e.g., [3]). Or does this warrant further investigation for future work? If yes, which part of the new formulation yields novelty compared to prior literature?**
> We thank the reviewer for this insightful comment and for pointing us towards Towards the Generalization of Multi-view Learning ([3]). And we really appreciate the connection drawn between our heuristic motivation and the formal generalization analysis in multi-view learning.
> While both works draw inspiration from the general Information Maximization principle, there are fundamental differences in the problem setting, formulation, and application that distinguish our work from [3] and make their specific bounds not directly applicable, though highly relevant for future theoretical extensions.
> We thank the reviewer for this thoughtful comment and for highlighting the connection to Towards the Generalization of Multi-view Learning [3]. While both works build on the Infomax principle, the theoretical setting in [3] is not directly applicable to ours due to key differences in problem setup and formulation:
>
> ### 1. Different learning regimes.
> [3] studies multi-view representation learning on static datasets, where views are fixed and sampled i.i.d. from a stationary distribution. In Self-Harmony, views are dynamically generated paraphrases during Test-Time RL, and the distribution of views changes as the Reframer policy adapts. This feedback loop breaks the i.i.d. assumption underlying the generalization bounds in [3].
>
> ### 2. Different Infomax objectives.
> [3] uses Infomax to disentangle shared vs. private latent representations. In contrast, our View-Invariant Infomax is defined over pseudo-label selection: maximize $I(Z_a; A)$ while minimizing dependence on the view $I(Z_a; X)$. The goal is not to learn a latent space but to identify stable answers under paraphrase transformations.
>
> ### 3. Novel Approximation.
> Our contribution is to convert this theoretical objective into a computationally efficient selector: the Harmonic Mean of answer frequencies. Unlike the entropic terms in [3], which require estimating complex mutual information, our proxy enables real-time scoring during inference.
>
> We've added this citation in revision.

---

> ### Author Response · Authors · 2025-11-22
> **Response to Reviewer v4TH (Part 5/5)**
>
> ## 10. **(Minor) Very recently, there have been some issues regarding floating-point operations (https://x.com/QPHutu/status/1984258808332550245). Can the authors comment on whether this issue(?) applies to their experiments?**
>
> The row MSE(fp16, bf16) reports, for each metric separately, the mean squared error between the fp16 and bf16 runs averaged across training steps. The differences are extremely small (2.02e-6, 1.53e-6, 2.63e-5), demonstrating that our training process and results are **stable across floating-point formats** and unaffected by the fp16 instability mentioned by the reviewer.
>
> |  | Critic Reward | Pseudo Label Accuracy | Mean of Pass@1 in 16 Rollouts |
> | --- | --- | --- | --- |
> | MSE(fp16,bf16) | 2.02e-6 | 1.53e-6 | 2.63e-5 |
>
> ## Conclusion
> We are grateful to the reviewer for the valuable questions which guided us to deepen both our theoretical explanation and experimental validation. The newly added analyses and derivations directly respond to the raised concerns and further reinforce the claims in the paper. We believe the revisions have substantially improved the manuscript and hope that the responses fully address the reviewer’s points. We have also incorporated the corresponding discussions into the revised manuscript.
>
> [1] https://www.jmlr.org/papers/v21/20-163.html
>
> [2] https://arxiv.org/abs/1907.02893
>
> [3] https://arxiv.org/abs/2501.16768

---

> > ### Comment · Reviewer_v4TH · 2025-11-26
> >
> > Thank you very much for providing comprehensive responses to all my concerns and questions, which have now been resolved. Accordingly, I raise my score and will champion for acceptance. Good luck!

---

> > > ### Author Response · Authors · 2025-11-26
> > > **Response to Reviewer v4TH**
> > >
> > > Thank you very much for your positive feedback. We are delighted that our responses have fully addressed your concerns. We sincerely appreciate your decision to raise the score and your willingness to champion our paper — your support means a great deal to us. Thank you again for the encouragement.

---

### Author Response · Authors · 2025-11-25
**General Response**

We sincerely thank all reviewers for their insightful feedback and constructive suggestions. We have carefully revised the manuscript in response to the questions and discussions raised during the rebuttal period. Below is a summary of the major modifications(in blue color):
1. Related Work — We expanded the related work section to include the papers highlighted by Reviewer v4TH and Reviewer RRhh.
2. Theorem 3.1 — We now explicitly state the assumption of a uniform prior and clarify the use of the second-order approximation, as suggested by Reviewer v4TH.
3. Section 3.1 — We revised the explanation of the connection between Liu et al. 2025b and the failure mode of majority voting/self-consistency, addressing the comment by Reviewer RRhh.
4. Section 4.1 — We added a detailed clarification on how each model is trained separately on each dataset/task, in response to Reviewer RRhh.
5. Section 4.3.1 — We corrected the citation placement for Table 2, based on the suggestion from Reviewer 962B.
6. Section 4.4 and Figure 5 — We included additional examples of reframed questions to better illustrate reframing behavior, as suggested by Reviewer RRhh.
7. Appendix E.3 — We added a sensitivity analysis on semantic shift in reframed questions, as suggested by Reviewer dk94.
8. Appendix E.4 — We added an explanation of how coefficients are assigned in the score function, in response to Reviewer v4TH.
9. Appendix G.2, G.3, and G.4 — We added theoretical derivations on following topic to address the theoretical questions raised by Reviewer v4TH.
    - Tunable λ and generalized mean
    - Multiple views and generalized harmonic mean
    - Non-uniform priors and weighted invariant score
10. Appendix H and Table 8 — We added a fine-grained comparison of different mean-based pseudo-label selectors against the TTRL baseline to address concerns raised by Reviewer v4TH, Reviewer dk94, Reviewer 962B, and Reviewer RRhh.
11. Appendix I and Table 9 — We added a inference-only comparison of self harmony against the TTRL baseline to address concerns raised by RRhh.
12. Appendix J and Table 10 — We added a comparative signal quality across confidence-correctness quadrants raised by dk94, RRhh.
13. Appendix K — We added a mathematical decomposition of the harmonic-mean formulation to clarify its underlying structure.
14. Appendix L and Figure 12 — We added a comparison of signal quality across training steps.

We sincerely appreciate the reviewers’ time, constructive insights, and efforts in helping us improve the quality and clarity of this work.

---

### Author Response · Authors · 2025-11-30
**Final Summary**

**Dear Area Chair:**

We respectfully submit this summary to assist with the final decision. Following our rebuttal, **Reviewer v4TH raised their score to 10 (Strong Accept)** and explicitly stated their willingness to **champion this paper** to acceptance.

While all reviewers **recognized multiple strengths of the work**, they also raised several targeted technical questions. These concerns have now been **systematically resolved** — through new experiments and theoretical extensions where appropriate, and through detailed clarifications for issues focused on related work and experimental setup.

---
## **1. Consensus Strengths**
1. Dominant Empirical Performance — All reviewers (v4TH, dk94, 962B, RRhh) acknowledged that Self-Harmony delivers superior results. Reviewer dk94 emphasized that the method ranks “1st in 28 of 30 settings” across six major reasoning benchmarks and five models in two model families.
2. Theoretical Grounding — Most Reviewers (v4TH, dk94, and RRhh) confirmed that the method is not a heuristic trick but is principledly grounded in the Infomax objective, offering a clear information-theoretic interpretation of pseudo-label selection.
3. Intuitive & Elegant Framework — The Dual-Role (Solver + Reframer) architecture was praised as “elegant, practical, and scalable” (dk94), “clear and solid” (v4TH), and important for enabling unsupervised signals in the wild (RRhh).
4. Robustness & Reproducibility — Reviewers noted the training stability (dk94) and robustness to hyperparameters (962B). v4TH praised the release of reproducible code.
5. High Quality of Presentation — dk94 highlighted that the paper “stands out for its clarity,” v4TH called it “well-written,” and RRhh described it as “well-structured and easy-to-follow.”
---
## **2. Factual Resolution of Major Technical Concerns**

Only the major issues are summarized below:

### Concern: What drives the gains beyond standard Self-Consistency / Majority Voting? (RRhh, 962B)

Evidence:
- Analytical Decomposition:
        $\text{HMS} = \frac{1}{2} \biggl[ \underbrace{{(p_{0}(a) + p_{1}(a))}}\_{\text{Self Consistency}} - \underbrace{{\frac{(p_{0}(a) - p_{1}(a))^2}{p_{0}(a) + p_{1}(a)}}}\_{\text{View Invariance}}\biggl]$. Self-Harmony suppresses answers that are confident but fragile.

- Universal Gains across Quadrants: The trained model improves Mean Pass@1 across all four quadrant subsets (confident-correct, confident-incorrect, not-confident-correct, not-confident-incorrect) — demonstrating benefits without trade-offs.
- Cleaner Training Signal (Correlation): On the Confident-Incorrect subset, Majority Voting produces noisy pseudo-labels (Spearman = 15.19%), whereas Self-Harmony provides a significantly stronger signal (Spearman = 36.51%).
- Inference-Only Utility: Even without training, Self-Harmony yields higher Pass@1 than Majority Voting across models (4B, 8B), rollout budgets (32–256), and all quadrant subsets.

Resolution: Self-Harmony enhances performance on both easy and hard subsets, with the largest gains occurring where Majority Voting fails, in both inference-only and training regimes.

### Concern:  What intuition and evidence support self-harmony's advantage over other aggregator? (v4TH, RRhh)
Evidence:
- A direct comparison against Arithmetic, Geometric, Quadratic Mean, and Infomax without second-order approximation

Resolution: Self-Harmony is consistently the strongest across all quadrants.
### Concern: Does paraphrasing drift from the Reframer incorrectly down-weight valid answers? (dk94)
Evidence:
- Sensitivity analysis varying both Reframer sizes (0.6B → 14B) and Solver sizes (1.7B → 8B) shows consistently large improvements over the baseline solver (before training) across all configurations. Even with a noisy 0.6B Reframer, the Confident-Correct subset improves sharply (like Qwen-4B: 83.98% → 96.48% in Mean Pass@1 under 16 rollouts).

Resolution: Self-Harmony does not penalize valid solutions under semantic drift and remains effective without requiring a strong Reframer.
### Concern: Can the theoretical framework be generalized? (v4TH)
Evidence:
- We show that introducing a tunable λ recovers different Generalized Means under second-order approximation.
- Theorem 3.1 was further extended to support multi-view and non-uniform prior assumptions.

Resolution: These theoretical extensions confirm that the framework is not restricted to the original two-view, uniform-prior, λ = 2 setting.

---
To conclude, we hope the following four points are helpful for the final decision:
1. State-of-the-art results in label-free test-time RL without external reward models.
2. A new pseudo-label selector that consistently outperforms majority voting in both inference-only and training settings.
3. Theoretical justification, grounding the method in Infomax with second-order extensions.
4. A simple and robust dual-role architecture that scales across models and reframer qualities, enabling stable label-free improvement of LLMs.

---

### Meta-Review · Area_Chair_2hh7 · 2025-12-13

**Summary:**

This paper proposes Self-Harmony, a test-time reinforcement learning framework that replaces majority voting with a view-consistency based pseudo-label selector using the harmonic mean across original and reframed questions.

Initial reviewer opinions were mixed. Reviewer v4TH rated the paper as accept and emphasized strong theory, clear intuition, and extensive experiments. Reviewer 962B gave a 6 and found the method sound and practical, though asked for deeper analysis of where gains come from. Reviewer dk94 rated it a 4, mainly due to concerns about the harmonic mean being heuristic and sensitivity to semantic drift. Reviewer RRhh rated it a 2, questioning the motivation, experimental rigor, and missing analysis relative to prior work on self-consistency and unsupervised reasoning signals.

The rebuttal is thorough. The authors added substantial new experiments, fine-grained analyses, and theoretical extensions. Most importantly, they directly addressed the core technical doubts: why harmonic mean works better than other aggregators, whether it fails under paraphrase drift, how it behaves on confident-incorrect cases, and whether it helps beyond majority voting even in high-consensus regimes. Reviewer v4TH explicitly raised their score to 10. After reading the paper and the discussion, I find the method simple but very well justified, and the empirical evidence convincing. The contribution feels solid and reliable for ICLR.

**Reviewer Concerns:**

Concerns that were addressed:

1. Why harmonic mean over majority voting or other means: The authors added detailed comparisons against arithmetic, geometric, quadratic means, and direct Infomax optimization, with consistent gains across confidence-correctness subsets.

2. Sensitivity to semantic drift in paraphrasing: New stress tests using very small Reframers show that performance improves even with noisy paraphrases, addressing dk94’s main concern.

3. Where the gains come from: The requested breakdown by confident vs non-confident and correct vs incorrect cases was added, with F1 and Spearman correlation, directly addressing 962B and RRhh.

4. Self-consistency baselines at high consensus: The authors showed that Self-Harmony still outperforms majority voting even when filtering to high-consensus cases.

5. Scalability and inference-only use: Additional results show gains across model sizes and even without any training updates.

6. Related work and theoretical clarity: Missing citations were added, assumptions were clarified, and the theory was extended to tunable lambda, multiple views, and non-uniform priors.

Remaining concerns:

1. RRhh remains skeptical about the framing relative to broader unsupervised reasoning literature, but the added experiments and metrics substantially weaken the original objections.

2. The approach is simple by design, which some may see as incremental, but this is balanced by strong empirical and theoretical support.

Overall, no major technical concerns remain unaddressed.

**Reviewer Scores:**

Reviewer v4TH: 8
Explicitly raised score after rebuttal and stated willingness to champion acceptance.

Reviewer dk94: 4
Their main concerns on harmonic mean justification and semantic drift were directly addressed with new experiments and analysis. Likely moves to marginal accept.

Reviewer 962B: 6
Requested analysis was fully added. Likely to strengthen their positive view.

Reviewer RRhh: 2
While still unconvinced stylistically, many technical objections were addressed with concrete evidence. A move to borderline reject or neutral seems reasonable, but not full acceptance.

Given the post-rebuttal trajectory, and the robustness of the added evidence, I recommend acceptance.

---

### Decision · Program_Chairs · 2026-01-26

Accept (Poster)